# Non-viral precision T cell receptor replacement for personalized cell therapy

Susan P. Foy[1,13 ✉], Kyle Jacoby[1,13], Daniela A. Bota[2,13], Theresa Hunter[1], Zheng Pan[1], Eric Stawiski[1], Yan Ma[1], William Lu[1], Songming Peng[1], Clifford L. Wang[1], Benjamin Yuen[1], Olivier Dalmas[1], Katharine Heeringa[1], Barbara Sennino[1], Andy Conroy[1], Michael T. Bethune[1], Ines Mende[1], William White[1], Monica Kukreja[1], Swetha Gunturu[1], Emily Humphrey[1], Adeel Hussaini[1], Duo An[1], Adam J. Litterman[1], Boi Bryant Quach[1], Alphonsus H. C. Ng[3], Yue Lu[3], Chad Smith[1], Katie M. Campbell[4], Daniel Anaya[1], Lindsey Skrdlant[1], Eva Yi-Hsuan Huang[1], Ventura Mendoza[1], Jyoti Mathur[1], Luke Dengler[1], Bhamini Purandare[1], Robert Moot[1], Michael C. Yi[1], Roel Funke[1], Alison Sibley[1], Todd Stallings-Schmitt[1], David Y. Oh[5], Bartosz Chmielowski[4,6], Mehrdad Abedi[7], Yuan Yuan[8], Jeffrey A. Sosman[9], Sylvia M. Lee[10], Adam J. Schoenfeld[11], David Baltimore[12], James R. Heath[3], Alex Franzusoff[1], Antoni Ribas[4,6,14 ✉], Arati V. Rao[1,14] & Stefanie J. Mandl[1,14 ✉]

T cell receptors (TCRs) enable T cells to specifically recognize mutations in cancer cells[1–3]. Here we developed a clinical-grade approach based on CRISPR–Cas9 non-viral precision genome-editing to simultaneously knockout the two endogenous TCR genes *TRAC* (which encodes TCRα) and *TRBC* (which encodes TCRβ). We also inserted into the *TRAC* locus two chains of a neoantigen-specific TCR (neoTCR) isolated from circulating T cells of patients. The neoTCRs were isolated using a personalized library of soluble predicted neoantigen–HLA capture reagents. Sixteen patients with different refractory solid cancers received up to three distinct neoTCR transgenic cell products. Each product expressed a patient-specific neoTCR and was administered in a cell-dose-escalation, first-in-human phase I clinical trial (NCT03970382). One patient had grade 1 cytokine release syndrome and one patient had grade 3 encephalitis. All participants had the expected side effects from the lymphodepleting chemotherapy. Five patients had stable disease and the other eleven had disease progression as the best response on the therapy. neoTCR transgenic T cells were detected in tumour biopsy samples after infusion at frequencies higher than the native TCRs before infusion. This study demonstrates the feasibility of isolating and cloning multiple TCRs that recognize mutational neoantigens. Moreover, simultaneous knockout of the endogenous TCR and knock-in of neoTCRs using single-step, non-viral precision genome-editing are achieved. The manufacture of neoTCR engineered T cells at clinical grade, the safety of infusing up to three gene-edited neoTCR T cell products and the ability of the transgenic T cells to traffic to the tumours of patients are also demonstrated.

The ultimate goal of any cancer therapy is to target and kill cancer cells while sparing normal cells. The human immune system is suited to achieve this goal owing to the specificity of TCRs. These receptors can distinguish single point mutations in the cancer genome that change the amino acid sequences of peptides presented by the major histocompatibility complex (MHC) on the cell surface of cancer cells[1–4]. Mutational neoantigens provide the main target for the activity of adoptive cell transfer (ACT) therapeutics involving tumour-infiltrating lymphocytes, antitumour T cells stimulated by immune checkpoint blockade or cancer-specific vaccines[3–10]. Development of a clinical-grade approach to efficiently isolate multiple TCRs would open a new way to potentially treat refractory cancers. Specifically, these TCRs would recognize mutated peptides presented by any of the six human leukocyte antigen (HLA) class I alleles in a patient and then be subsequently engineered back into autologous T cells for ACT therapy. However, this goal has been limited by the diversity of HLA class I alleles in the human population, with over 24,000 alleles currently recorded[11]. Moreover, the polymorphic nature of most of the mutational antigenic determinants recognized by T cells is a challenge[3]. These are some of the reasons why most current TCR-engineered T cell therapies are limited to patients with the HLA-A:02*01 haplotype[12]. The generation of arrays of mutated peptide–HLA TCR-binding reagents and single-cell cloning of the paired TCR chains provided a new approach to isolate neoTCRs across multiple HLA alleles[13]. Furthermore, previous engineering of human T cells has

relied on the use of recombinant viral vectors[12,14], but it is unfeasible to generate multiple personalized clinical-grade vectors for every patient. The advent of nuclease-based precision gene-editing has enabled the development of approaches that use targeted insertion of transgenes into human T cells[15] and has paved the way to achieve stable integration without requiring the use of viral vectors[16]. Precision gene-editing using techniques such as CRISPR–Cas9 enables the simultaneous knockout of endogenous TCR chains while inserting the transgenic TCR under the control of the physiological TCR promoter. This strategy has been reported in some settings[17,18], but not all[19], to provide advantages over the same transgenes expressed under the control of constitutive viral vector promoters.

In the current study, we describe an approach that enables efficient isolation of multiple TCRs specific for mutational neoantigens. This strategy uses personalized libraries of hundreds of predicted neoantigen peptide sequences presented by HLA class I alleles of an individual patient and a targeted, non-viral gene-editing method to reconstitute the specificity of the isolated neoTCRs. Overall, this time-efficient process is able to generate clinical-grade neoTCR transgenic T cell preparations for ACT (Fig. 1a). Using these technologies, 16 patients with different solid cancers received up to three unique lots of gene-engineered T cells. Notably, these cells express patient-specific neoTCRs that target specific mutations of their cancer.

## Personalized isolation of neoTCRs

Patients provided consent to undergo a tumour biopsy and donate peripheral blood mononuclear cells (PBMCs) as part of the screening step for the selection of personalized neoTCR products (Extended Data Fig. 1a). Germline DNA from PBMCs was compared with tumour DNA using whole exome sequencing (WES) for the identification of patient-specific tumour mutations. In addition, RNA sequencing (RNA-seq) was performed to determine the expression level of genes with cancer-specific mutations. For the 16 patients who received treatment in the clinical trial, a median of 102 (range of 31–488) non-synonymous somatic mutations (NSMs) and a median of 35 (range of 20–236) expressed mutations were identified for each patient (Extended Data Table 1). Using this information, we prioritized a list of up to 352 neoantigen peptide–HLA candidates per patient (median of 352, range of 86–352, for a total of 5,302 peptide–HLA candidates with peptide lengths of 8–11 amino acids) across the available HLA class I alleles for that patient. Libraries of peptide–HLA complexes were produced in 293 cells as single-chain trimers, for which neoantigen peptides were fused in sequence to $\beta_2$ microglobulin (B2M) domains and HLA. Of these, a median of 104 (range of 49–262) patient-specific peptide–HLA proteins were successfully synthesized (yield of ≥2.7 µg) per patient for a total of 1,841 peptide–HLA proteins produced (Extended Data Table 1). These covered a total of 34 unique HLAs of the 64 HLA alleles represented in the screening process (Supplementary Table 1), with a median of 5 (range of 2–6) HLA alleles covered per patient (Extended Data Table 1). The peptide–HLA proteins were then DNA barcoded, fluorescently labelled and multimerized. CD8 T cells enriched from PBMCs of the patient were then stained using the patient-specific peptide–HLA multimer library to isolate rare peripheral blood-circulating T cells for any of the predicted mutations and HLA complexes. Single-cell sorting with high sensitivity and specificity was then performed. T cell isolation included CD95 cell surface staining to exclude CD8 T cells with a naive phenotype. Using this approach, we identified a median of 8 (range of 3–30) specific TCRs from T cells of varying frequency per patient, which resulted in a total of 175 TCRs for the 16 patients in the clinical trial. These TCRs recognized a median of 5 (range of 3–11) unique NSMs for a total of 83 neoantigen mutations (examples of typical data readouts are given in Fig. 1b,c).

## neoTCR product selection

The biological and potential therapeutic relevance of the captured candidate TCRs against the tumour neoantigens was then corroborated using precision genome-engineering of healthy donor T cells with each of the patient-isolated neoTCR candidates. The TRAC and TRBC genes from each captured T cell were single-cell sequenced and cloned into homologous recombination (HR) DNA plasmids. These HR plasmids were used together with site-specific nucleases to knockout the endogenous TCRβ chain and insert the transgenic TRAC and TRBC genes into the endogenous TRAC locus of CD8 and CD4 T cells (Fig. 2a). Each neoTCR candidate was then tested for recognition of the cognate neoantigen or the mismatched neoantigen controls to confirm neoantigen peptide–HLA-specific binding and interferon-γ (IFNγ) cytokine secretion. Binding of the reconstituted neoTCR to the soluble peptide–HLA complex demonstrated that 73 out of the 127 tested TCRs (57%) were specific and sufficiently functional neoTCRs for product selection (Extended Data Table 1). This result highlights the value of testing candidate neoTCRs before proceeding with clinical-grade manufacturing of the neoTCR gene-modified T cell product. Preclinical proof-of-concept data demonstrated that the neoTCRs isolated using our approach and then engineered into primary T cells were able to specifically recognize and kill tumour cells expressing endogenous levels of the neoantigen[20] (Extended Data Fig. 1b). Furthermore, we generated T cell products for seven clinically active TCRs[21–27] to retrospectively benchmark the selected neoTCRs for use in our trial. These results showed that approximately 50% (18 out of 37) of the neoTCRs we selected have a similar TCR potency as measured by the levels of IFNγ production (Extended Data Fig. 1c).

Up to three confirmed neoTCR candidates per patient were selected for clinical manufacture. We focused on TCR functionality and TCR binding in CD8 and CD4 T cells, diversifying among HLAs and the targeted neoantigens and clonality of the targeted neoantigen (Extended Data Table 2). The resulting 37 neoTCRs that were infused into the 16 patients in this clinical trial had IFNγ half maximal effective concentration (EC$_{50}$) values between 0.4 pg ml$^{-1}$ for the highest affinity TCR to 362 pg ml$^{-1}$ for the lowest affinity TCR. Eighteen of the neoTCRs (48%) had an EC$_{50}$ value greater than 30 pg ml$^{-1}$ and 11 (30%) had an EC$_{50}$ value between 30 and 3 pg ml$^{-1}$, and these were considered good-affinity TCRs. Meanwhile, 8 (22%) had an EC$_{50}$ value lower than 3 pg ml$^{-1}$, and these were considered high-affinity TCRs (Extended Data Table 2). The full-length TCR sequences, HLA alleles and neoepitope sequences for the 37 TCRs are provided in Supplementary Table 2.

## Non-viral precision TCR replacement

Patients with relapsed or refractory metastatic solid tumours and the following criteria were considered for enrolment: disease progression with standard-of-care treatment options for their cancer; adequate performance status; and evidence of measurable disease and fulfilling eligibility criteria. These participants were considered for leukapheresis when up to three neoTCRs had been selected using the personalized prediction and isolation of neoTCRs from their PBMCs and tumour biopsy samples. Patients underwent leukapheresis at their local institution, and the leukapheresis product was shipped to the sponsor (PACT Pharma) for manufacture. CD4 and CD8 T cells were isolated using automated magnetic separation cell sorting and activated for 2 days. Next, cells were electroporated to introduce Cas9 protein, guide RNAs to knockout the endogenous TRAC and TRBC genes and a HR template plasmid encoding the transgenic neoTCR. T cells were placed back in culture for expansion for 11 days, and the final cell product was cryopreserved on day 13 for infusion on a flexible schedule (Fig. 1a).

The single-step precision genome-engineering process results in the replacement of the endogenous TCR with the patient's native sequence neoTCR, the expression of which is placed under the regulation of the endogenous promoter (Fig. 2a). Infused CD8 and CD4 T cells expressing

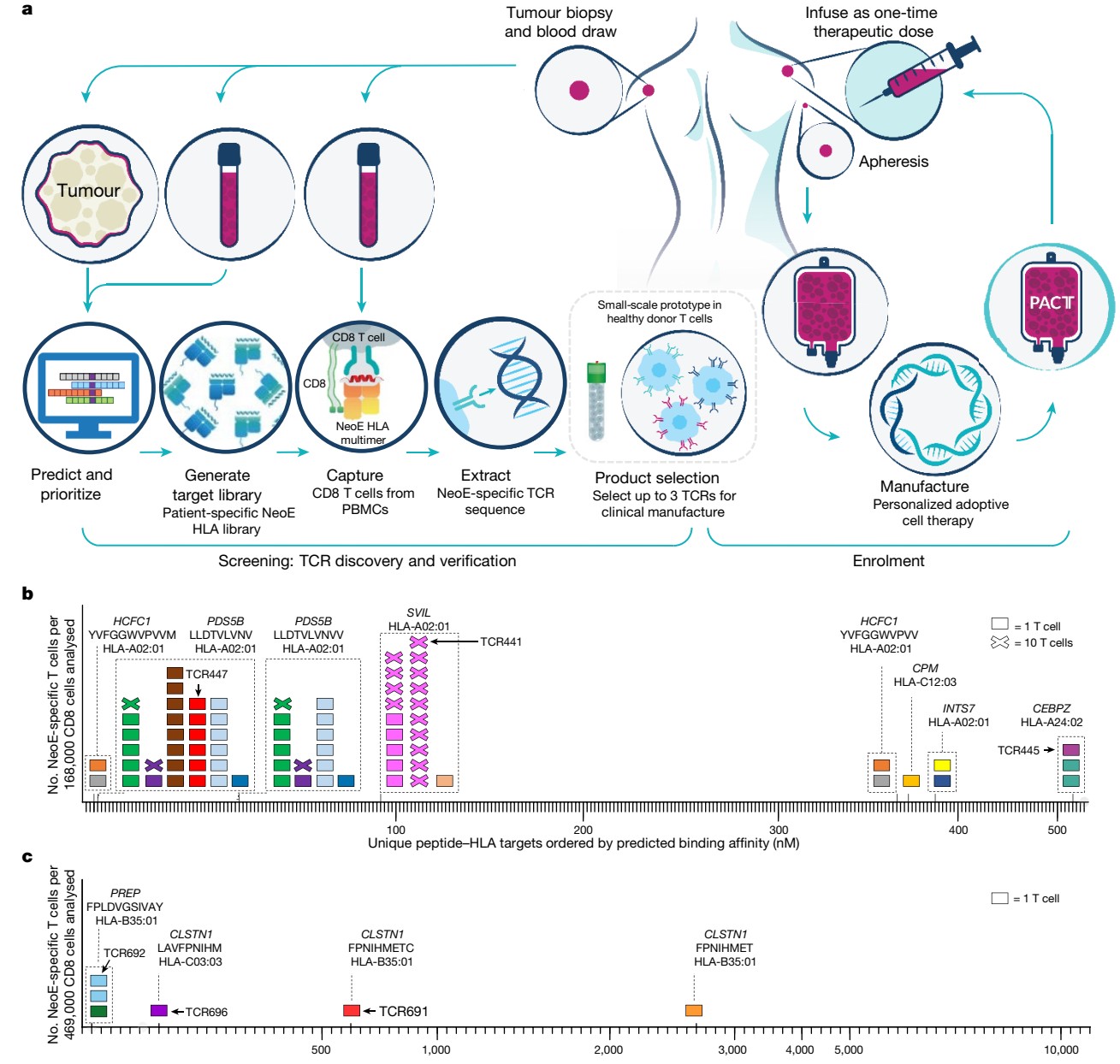

**Fig. 1 | Schematic for TCR discovery to cell manufacture. a,** Generation of the neoTCR product for each patient is separated into two steps: screening and enrolment. Screening begins with the identification of patient tumour-specific mutations based on sequencing data and bioinformatics prediction of mutated neoantigen peptides. Native CD8 T cells that bind neoantigen targets are captured from blood using barcoded and fluorescently labelled peptide–HLA multimers. neoTCR sequences are cloned from captured T cells and functionally characterized in healthy donor T cells before product selection. HR DNA plasmid (or plasmids) encoding the selected neoTCR sequence (or sequences) are then manufactured for subsequent clinical GMP T cell manufacture. Patients are enrolled into the study after product selection. Manufacturing begins with apheresis of the patient's blood followed by enrichment of CD8 and CD4 T cells. T cells are precision genome-engineered ex vivo to express one neoTCR. Cells are expanded and cryopreserved until the patient is ready for infusion. **b,c,** Two examples of neoTCR T cells isolated from patient PBMCs. Each box represents one T cell (x = 10 T cells), and each colour represents a TCR clone. T cells within dashed boxes target the same peptide–HLA target. Neoepitope (neoE) amino acid sequences and restricting HLA alleles are indicated on top of the boxes. Peptide–HLA targets are indicated by tick marks. An upward x axis tick indicates peptide–HLA that bound to a patient's TCR. All T cells shown on the graphs were a non-naive phenotype based on CD95 expression. TCRs indicated with numbers and arrows were selected for clinical-scale manufacture. **b,** Patient 0010. A total of 262 peptide–HLAs were made, and 236 neoantigen-specific T cells were isolated, representing 15 unique neoTCRs. The neoTCRs targeted six neoantigens across three HLAs. **c,** Patient 0506. A total of 105 bar-coded peptide–HLAs were made, and 6 neoantigen-specific T cells were captured, representing 5 unique neoTCRs. The neoTCRs targeted three neoantigens across two HLAs.

the neoTCRs were detected by flow cytometry analysis using the cognate fluorescently labelled peptide–HLA multimer. These cells constituted 1.9–46.8% of the live cell product (Extended Data Table 2). The remaining cell product consisted of T cells that had the endogenous TCR knocked out but no knock-in of the neoTCR or wild-type T cells still expressing the endogenous TCR. Relative neoTCR affinity was determined by the ability of the cognate peptide–HLA multimer to bind the neoTCR expressed by CD8 or CD4 T cells. Higher affinity neoTCRs can bind the peptide–HLA target in the absence of the CD8 co-receptor, whereas lower affinity TCRs need CD8 co-receptor stabilization. On the

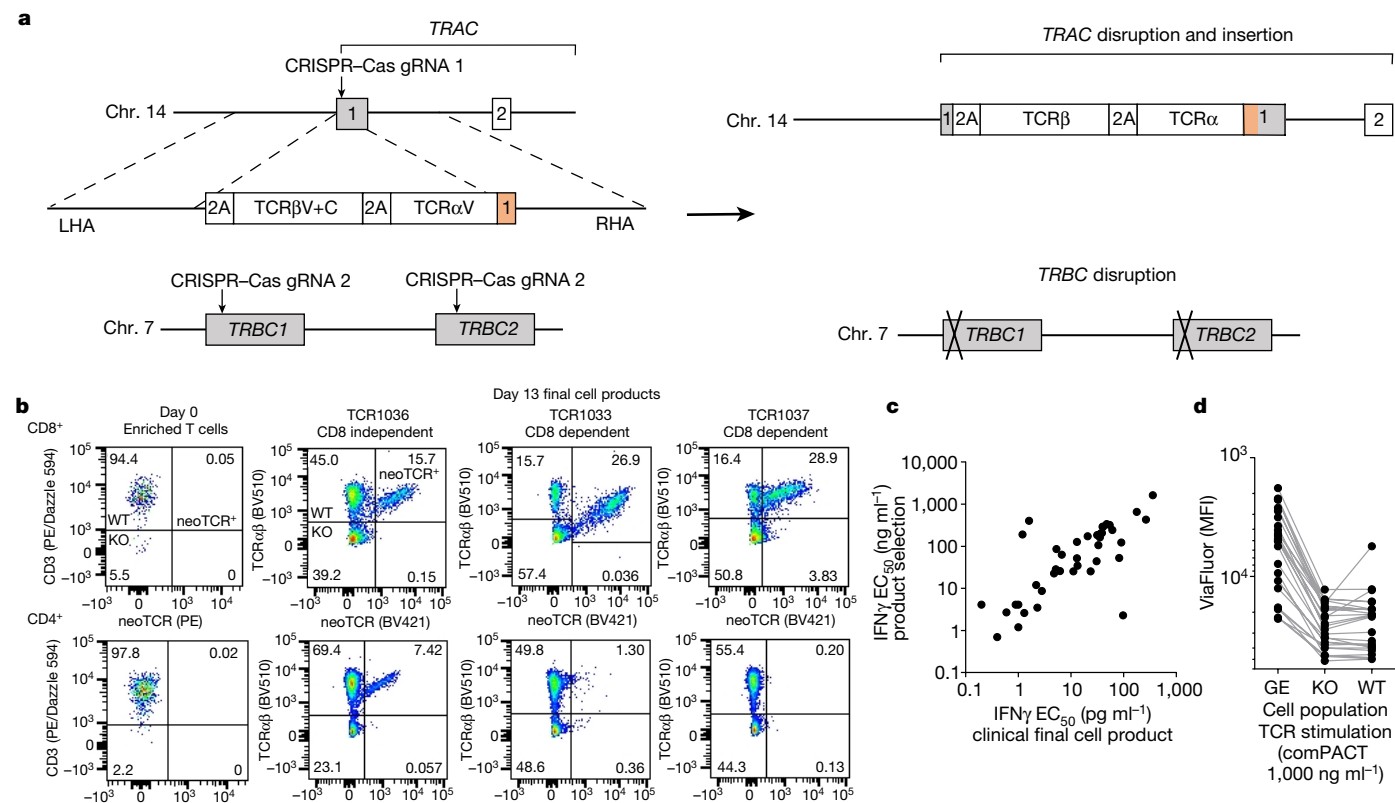

**Fig. 2 | Non-viral precision genome-engineering for clinical-grade cell manufacture. a**, Schematic of the construct design and resulting editing. **b**, Examples of endogenous TCR knockout and knock-in of up to three neoTCRs in a final clinical-grade cell product. Day 0 (left column) shows an example of the same patient's enriched T cell product but was not stained with the peptide–HLA multimer. Day 13 flow plots (right 3 columns) show the results of each of the 3 neoTCR product lots for that patient. **c**, TCR functionality (potency) as evaluated by IFNγ production correlates between small-scale products generated from healthy donor T cells and the final large-scale clinical-grade cell product. The functionality of the neoTCR clinical-grade product made for

patient autologous cells (IFNγ $EC_{50}$ values measured by ELISA or ELLA Simple Plex) was correlated with the functionality of the neoTCR product made in healthy donor cells at product selection (IFNγ $EC_{50}$ values measured by CBA; Pearson's $r = 0.8412$, $P < 0.0001$). **d**, Proliferation analysis of the final neoTCR clinical-grade cell product following exposure to peptide–HLA stimulation at 1,000 ng ml$^{-1}$. Each dot represents a unique neoTCR product. KO, knockout of the endogenous TCR only, these cells do not have a TCR on their surface; MFI, mean fluorescence intensity; neoTCR$^+$ or GE, gene-edited knockout of wild-type TCR and knock-in of neoTCR; WT, wild-type, unedited cells expressing the endogenous TCR.

basis of this information, TCRs can be classified as CD8-dependent or CD8-independent binders (Fig. 2b). Both CD8 and CD4 T cells showed insertion of the transgenes, as detected by intracellular staining for the 2A peptide that separates the neoTCR α-chain and the β-chain (Fig. 2b and Extended Data Fig. 1d). Assessment of the clinical cell products using targeted locus amplification confirmed on-target integration of the transgenic TCR cassette (Extended Data Fig. 1e). Fluorescence in situ hybridization (FISH) analyses indicated a slight increase ($P = 0.0137$) in chromosomal aberrations at the target sites of chromosomes 7 and 14 (Extended Data Fig. 1f), which indicated the presence of *TRAC:TRBC* translocations. However, other studies[28,29] have shown that cells harbouring translocations, including cells with *TRAC:TRBC* translocations, do not exhibit a growth advantage, and the frequency of these translocations decreased over time.

The potency of the final cell product was assessed through ELISAs to measure IFNγ secretion from neoTCR gene-edited T cells exposed to the cognate peptide–HLA multimer. The results were highly correlated (Pearson's $r = 0.8412$, $P < 0.0001$) with IFNγ responses measured by cytokine bead array (CBA) of the same neoTCR in healthy donor cells generated at small-scale for initial product selection (Fig. 2c). These data provide further validation of the neoTCR product selection process. In addition to IFNγ secretion, cells displayed a polyfunctional cytokine secretion profile in response to mutational neoantigen peptide–HLA complexes, expressing CD107a on the cell surface and producing

tumour necrosis factor (TNF) and interleukin-2 (IL-2) (Extended Data Fig. 2a). neoTCR T cells proliferated in a dose-dependent manner in response to neoantigen peptide-HLA stimulation (Fig. 2d and Extended Data Fig. 2b,c). Wild-type cells (defined as having no evidence of CRISPR knockout or knock-in of the TCR) proliferated in response to positive control T cell stimulation using anti-CD3 and anti-CD28, but not with the cognate neoantigen peptide–HLA complexes. Cells in which *TRAC* and *TRBC* are knocked out do not have a TCR in complex with CD3 chains on their surface. Consequently, these cells did not proliferate in response to stimulation with either the anti-CD3 plus anti-CD28 positive control stimulation or with cognate neoantigen peptide–HLA-specific stimulation. Additional characterization revealed that the T cell phenotype shifted from a predominantly T effector-like cell phenotype in the incoming leukapheresis product to a predominantly T memory stem cell and T central memory cell phenotype for each lot of neoTCR cell product (Extended Data Fig. 3a). There was infrequent expression of the co-stimulatory and co-inhibitory surface markers 4-1BB (also known as CD137), LAG-3 and PD-1 (Extended Data Fig. 3b). There was frequent expression of TIM-3, which has been previously related to the presence of γ-chain cytokines in cell culture[30]. CD73 and CD39, which are upregulated on T cells during activation and differentiation[31], were also expressed by the manufactured cell product.

During the course of the clinical trial, the formulation of the culture medium was changed to improve cell growth and editing efficiency of

the clinical-grade preparations (cell manufacture version 2.0 (v.2.0) to v.2.1), and we also changed the electroporation device (cell manufacture v.2.1 to v.3.0; Supplementary Table 3). The change in culture medium affected the quality and quantity of the clinical-grade preparations, with an increase in neoTCR knock-in efficiency from 13.4% to 23.0% (range of 1.9–28.3% and 11.4–46.8% for cell manufacture v.2.0 and v.2.1, respectively) and increases in the total number of neoTCR transgenic T cells manufactured ($1.08 \times 10^9$ to $1.78 \times 10^9$). The switch in electroporation devices improved the knock-in and knockout efficiencies, reducing the wild-type population in the final product to <10% (Extended Data Table 2 and Extended Data Fig. 3c, d).

## Patient enrolment and treatment delivery

From December 2019 to August 2022, the study was active at 9 investigational sites. A total of 187 patients provided informed consent to initiate the process of personalized neoTCR discovery. After successfully obtaining the appropriate quality of baseline biopsy samples and PBMCs for DNA sequencing and RNA-seq, 88 patients (47%) participated in the neoTCR discovery process. Of these participants, 46 (52%) met the requirements for successful product selection for clinical manufacture (Fig. 3a). Twenty-eight patients underwent apheresis, and data from 16 patients who received the neoTCR transgenic T cell products produced using the v.2.0, v.2.1 or v.3.0 manufacturing process are reported here. Patients had a median age of 47 years (range of 36–70 years) and had received a median of 5 previous lines of therapy (range of 2–9) at the time of consent. The following number of patients with cancer histologies across the tumour mutation burden spectrum were enrolled: 11 patients with microsatellite stable-colorectal cancer; 2 with hormone-receptor-positive breast cancer; 1 with ovarian cancer; 1 with melanoma; and 1 with lung cancer (Extended Data Table 3). Patients received a total of $4 \times 10^8$ (dose level 1), $1.3 \times 10^9$ (dose level 2) or $4 \times 10^9$ (dose level 3) neoTCR transgenic T cells within 3 dose cohorts. An additional four patients who received the first two dose levels also received subcutaneous IL-2 after infusion. Per the protocol, IL-2 could be added to the regimen once at least three patients had received that dosage regimen and demonstrated safety. Nine patients received cell products with three neoTCRs, three with two neoTCRs and four with one neoTCR each. The four patients who were assigned to the dose level 1 group received a lymphodepleting conditioning regimen of 300 mg m$^{-2}$ of cyclophosphamide and 30 mg m$^{-2}$ of fludarabine for 3 days. This led to suboptimal depletion on the basis of absolute lymphocyte counts. The regimen was modified for the remainder of the study and was increased to 600 mg m$^{-2}$ of cyclophosphamide for 3 days and 30 mg m$^{-2}$ of fludarabine for 4 days (Extended Data Table 3 and Extended Data Fig. 4a). A step-by-step timeline from TCR discovery and validation to product release for each patient is shown in Extended Data Fig. 4b. The screening step to TCR discovery took a median of 167 days, which included the following steps: provision of informed consent and acceptance of the sequenced biopsy sample for pipeline analysis (60 days); bioinformatics (11 days); isolation of neoantigen peptide–HLA complexes (62 days); neoTCR functional characterization in healthy donor T cells (29 days); and product selection of the neoTCRs for dosing (5 days). After selection of the neoTCRs, patients were eligible for enrolment into the clinical trial, with a median time of 102 days between enrolment and dosing. This time period included the manufacture of plasmids (median 11 days) and neoTCR transgenic T cells (13 days), and testing of the lot released (28 days). It also included additional time before apheresis was scheduled (32 days). Once a product passed lot release, there was a median of 18 days before the patient received the dose of neoTCR transgenic T cells.

## Infusion of neoTCR-engineered T cells into patients

We performed repeated peripheral blood sampling from patients to study the engraftment and expansion of the neoTCR transgenic T cells

after infusion over time. The peak of total neoTCR transgenic T cells detected in blood samples from patients increased with the increase in cell dose. The increase occurred at a median time of 2 days after infusion (range of 1–15 days) for the 16 patients who received treatment. Moreover, the per cent of neoTCR$^+$ cells in the infused product correlated with the pharmacokinetic area under the curve up to day 7 (Fig. 3b and Extended Data Fig. 4c,d). For dose level 1, with the original lymphodepleting regimen, the median peak of neoTCR transgenic cells was 1.9% (range of 0.9–3.3%), which increased to a median of 10.0% (range of 7.7–12.1%) and 15.0% (range of 12.0–37.7%) for dose level 2 and level 3, respectively, in patients who received the modified conditioning chemotherapy regimen. With the original conditioning chemotherapy regimen, patients had absolute lymphocyte counts of fewer than 100 cells per µl for a median of 1.5 days (range of 0–3). This increased to a median of 4 days (range of 1–10) with the modified conditioning regimen (Extended Data Fig. 4a). The addition of subcutaneous IL-2 resulted in a peak of neoTCR transgenic T cells of 7.3% and 6.3% in two patients in the dose level 1 group (patients 0604 and 0411, respectively). Patient 0026 (assigned to the dose level 2 group and receiving cells manufactured using v.2.0 of the process) received IL-2 and had a peak of 9.5% neoTCR transgenic T cells. Two patients who received cell products generated using v.3.0 of the manufacturing process had the highest levels of circulating neoTCR transgenic T cells, peaking at 20.8% for patient 1003 with dose level 2 and the addition of subcutaneous IL-2, and at 37.3% for patient 0417 with dose level 3. Overall, increasing the chemotherapy concentration in the lymphodepleting regimen, adding IL-2 and increasing the number of neoTCR$^+$ cells while decreasing the number of wild-type cells in the final cell product each may have contributed to improving the maximum concentration and prolonging the exposure of neoTCR T cells detected in the peripheral blood (Fig. 3b and Extended Data Fig. 4a,c,d).

One role of the lymphodepleting conditioning regimen is to increase the availability of the T cell homeostatic expansion cytokines IL-7 and IL-15 in serum, which favours the expansion of the infused T cells[14]. Increased IL-7 and IL-15 levels were observed in serum from patients in the clinical trial, with peak concentrations primarily observed at the time of infusion (Extended Data Fig. 5a). Analyses of other circulating cytokines demonstrated early peak increases in IFNγ, TNF, IL-6, IL-8 and IL-10 in some patients. There was no clear relationship between these levels and toxicities, and the levels were much lower than serum levels of these cytokines observed in cases of cytokine release syndrome in other clinical trials[32,33]. Circulating IL-2 could only be detected in the patients who received supplemental subcutaneous IL-2 (Extended Data Fig. 5a).

neoTCR transgenic T cells recovered from blood after infusion generally maintained the phenotypes of the infusion product, and T memory stem cells and T effector-like cells were the predominant phenotypes (Extended Data Fig. 5b). There were infrequent increases in the surface expression of CD73, LAG-3 and PD-1, whereas CD137 remained low (Extended Data Fig. 5c). Expression of TIM-3 and CD39 were decreased in the neoTCR transgenic T cells recovered from patients after infusion compared with before infusion. This result indicates that the expression of these factors in the infusion product was due to activation on day 2 and the presence of exogenous cytokines during the manufacturing process[30].

## neoTCR T cells in tumour biopsy samples after treatment

In addition to the biopsy sample provided at screening for neoantigen prediction and identification, patients provided a baseline biopsy sample (day 5–7 before lymphodepletion) to assess continued neoantigen gene expression, and a biopsy sample after infusion (day 5–7 or day 28) to analyse for neoTCR transgenic T cell infiltration. Of the eight patients with longitudinal biopsy samples available for analyses, 13 out of the 19 predicted truncal neoantigen targets were detected from biopsy

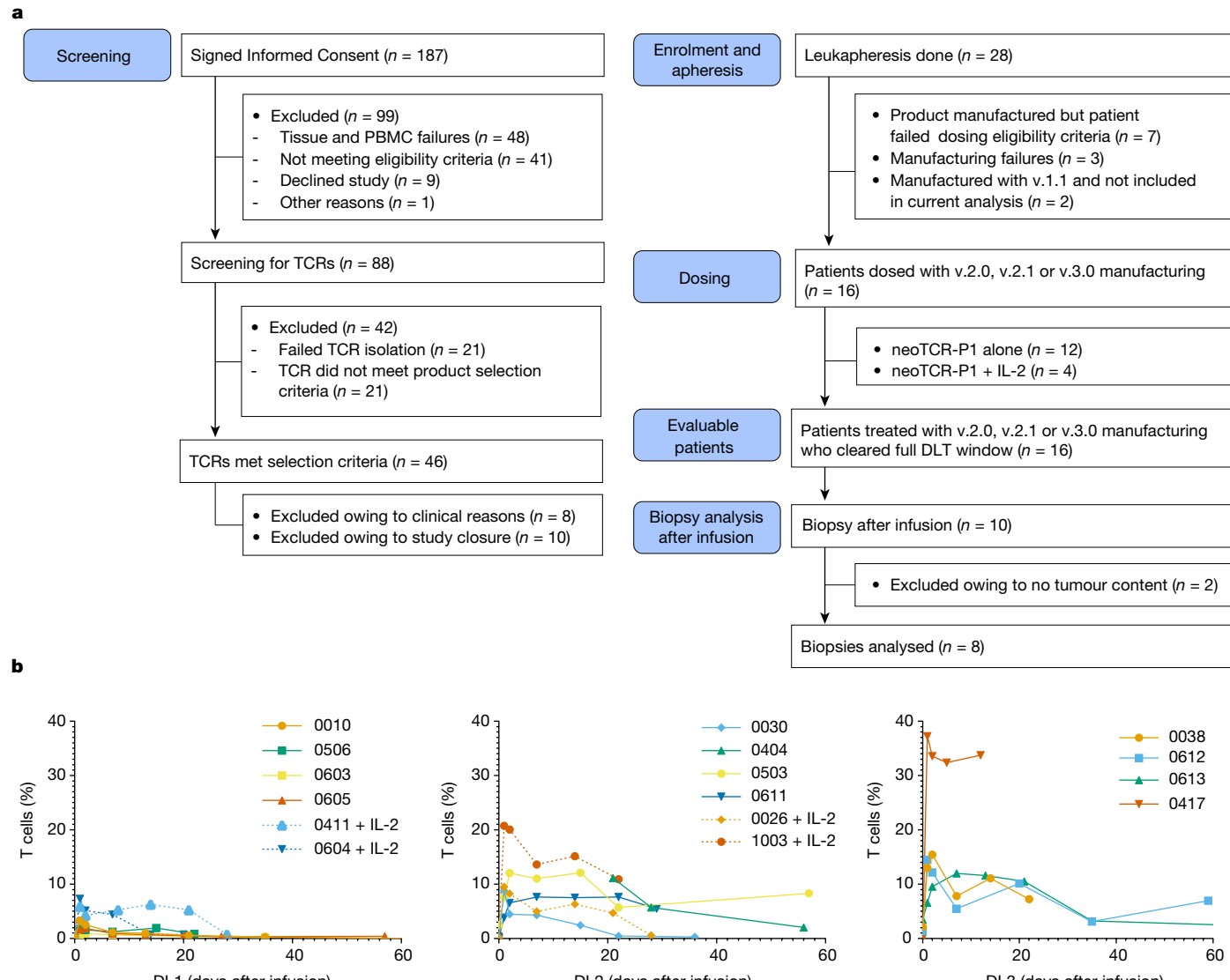

**Fig. 3 | Clinical trial patients and samples, and analysis of neoTCR transgenic T cells in blood after infusion. a**, Consolidated standards of reporting trials diagram, with the number of patients who provided consent, continued onto TCR isolation and leukapheresis, had clinical products manufactured for them, were infused with neoTCR transgenic T cells and provided blood and biopsy samples for analyses. DLT, dose-limiting toxicity. **b**, Expansion and persistence of neoTCR transgenic T cells in peripheral blood of patients, as measured by flow cytometry of stained peptide–HLA multimer cells. Percentages of total T cells from patients in the dose level 1 (DL1), DL2 and DL3 groups are shown. Patients who also received IL-2 therapy are indicated by dotted lines. All available time points were analysed. For patient 0613, 1.3% neoTCR⁺ cells was measured at day 106 after infusion. The limit of detection is approximately 0.16%.

samples provided at the screening and baseline steps (Extended Data Fig. 6a). Patient 0010 was the only patient whose targeted mutations were not detected in the longitudinal baseline sample or the sample after infusion (*n* = 3). This result can be explained by this patient having a cancer with a strong APOBEC somatic mutation signature (retrospective analysis; Extended Data Fig. 6b), which has previously been reported to drive extreme tumour heterogeneity[34]. Patient 0605 had a neoTCR targeting a neoepitope that was predicted to be a subclonal mutation in *GPSM2*, which was absent in follow-up biopsy samples. Patient 0506 had a neoepitope mutation in *PREP* that was undetectable in longitudinal tumour biopsy samples but was detectable using a bespoke circulating tumour DNA (ctDNA) assay at both subsequent time points (Extended Data Fig. 6c). This result suggests that ctDNA may be complimentary to tissue biopsy samples for the accurate identification of subclonal mutations. In addition, we performed retrospective HLA loss of heterozygosity (LOH) analyses on biopsy samples from

patients who received the infusions. The analyses showed that LOH of the specific HLA allele presenting a selected neoantigen epitope had occurred before treatment in 3 patients, which affected 4 out of the 37 TCRs that were selected and infused (Extended Data Fig. 7a). Hence, a subset of the neoantigen mutations presented by specific HLA alleles that had been targeted by the dosed neoTCRs were not presented at the time of infusion. This finding demonstrates both the importance of HLA LOH analysis for TCR T cell therapies and the value of identifying sets of therapeutic TCRs that are not limited by specificity against neoantigens presented by a single HLA allele.

Sequence quantification of the TCR complementary-determining region 3 (CDR3) was performed on tumour biopsy samples taken before and after infusion (Fig. 4a,b and Extended Data Fig. 7b). The genetically introduced neoTCRs were frequently among the top represented TCR CDR3 sequences in these biopsy samples. In detail, 12 of the infused neoTCR sequences were among the top 4% CDR3 sequences found in

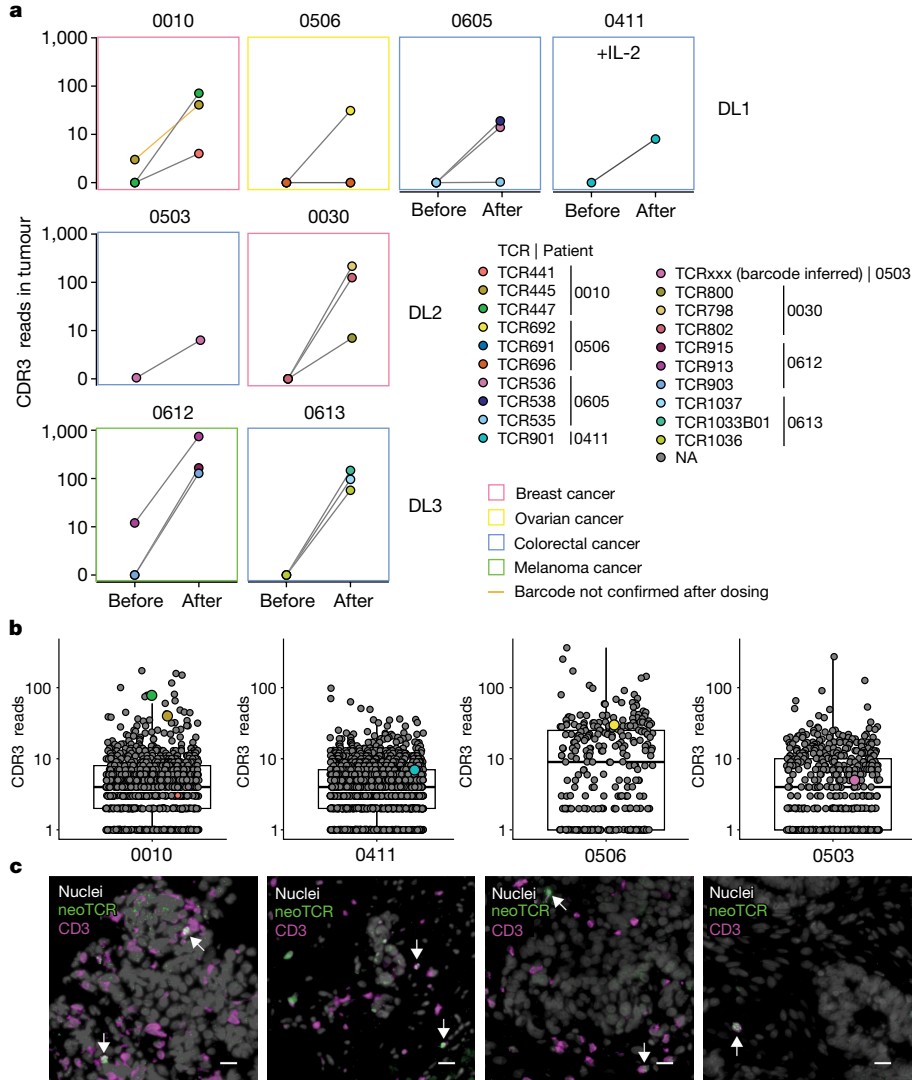

**Fig. 4 | Trafficking of neoTCR transgenic T cells as detected in tumour biopsies. a**, Analysis of all available biopsy samples at baseline (Before) and after infusion (After) for the presence of infused neoTCRs from detected CDR3α and barcoded reads for DL1, DL2 and DL3 with or without IL-2. Colours of the boxes indicate cancer type. One TCR for which the sequencing reads supporting the codon-optimized constant region was not detected is marked with an orange line (patient 0010). NA, not applicable. **b**, Quantitation of TCR CDR3α reads in biopsy samples from patients after infusion. Dots are labelled both by size (larger if the neoTCR barcode was confirmed) and colour. Coloured dots are CDR3 sequences matching neoTCRs, and grey dots are neoTCR-unrelated CDR3α reads and their relative quantification. Boxes indicate the interquartile range (IQR), the centre line the median, and the whiskers the lowest and highest values within 1.5× the IQR from the first and third quartiles, respectively. **c**, Spatial profiling of four patients to image neoTCR transgenic T cells in tumours after treatment. Grey, nuclei; green, neoTCR; magenta, CD3. White arrows denote neoTCR T cells. Scale bar, 20 μm.

the samples after infusion, 6 of which were from patients who were in the dose level 3 group (Fig. 4b and Extended Data Fig. 7b). Immediately flanking the CDR3 sequence is a short codon-optimized constant region that can serve as a barcode to differentiate between the native neoTCR and the transgenic neoTCR CDR3 sequences (Extended Data Fig. 7c). This enabled us to distinguish the neoantigen-specific native T cells from neoTCR transgenic T cells and to quantify the infiltration of neoTCR transgenic T cells into tumours after infusion. Using this approach, 22 separate populations of infused neoTCR transgenic T cells were detected in the tumours. Notably, only 2 out of the 22 populations (TCR913 and TCR441) of neoantigen-specific native T cells were evident in pre-infusion tumour biopsy samples. TCR441 was present at a high frequency in the blood at the time of product selection for patient 0010 (145 per 168,000 CD8 cells; Fig. 1b), constituting 0.09% of CD8⁺ T cells in the blood. In both the biopsy samples taken at baseline and after infusion, the TCR441 barcode could not be detected directly. However, the native sequence was detected, which suggests the presence of native

but not transgenic TCR441. TCR913 (from patient 0612) was detected in the pre-infusion and after infusion biopsy samples, and detection of the TCR913 barcode in the sample after infusion suggests the presence of the infused TCR913 transgenic T cells. TCRs with a lower IFNγ $EC_{50}$ value or higher affinity score (a measure of CD8-independent binding) were more likely to be found in the tumour. Specifically, 16 out of the 22 individual T cell populations were detected in the analysed biopsy samples taken after infusion from the 8 patients with samples available for analyses (Extended data Fig. 7d; *$P < 0.05$). For patient 0503, who received an infusion containing three neoTCR products, the presence of the neoTCRs was inferred by the flanking barcode sequence (Fig. 4a,b), but the CDR3 sequence for the specific neoTCR could not be resolved.

The presence of neoTCR transgenic T cells in the biopsy samples after infusion was additionally confirmed by fluorescence microscopy using RNAscope in situ hybridization (ISH) probes that were developed to specifically detect the mRNA sequences of the neoTCR knock-in construct. Using this approach for four of the biopsy samples taken

after dosing, we directly visualized the intratumoural presence of the neoTCR transgenic T cells. These cells were frequently in contact with cancer cells (Fig. 4c), which demonstrates that the infused neoTCR transgenic T cells trafficked to solid tumour metastases. The neoTCR transgenic T cell frequency determined from these images showed good correlation with the neoTCR transgene sequencing read counts (Pearson's coefficient of 0.8; Extended Data Fig. 7e).

## Toxicities, responses and patient outcomes

All patients experienced grade 3–4 pancytopaenia, which is an expected toxicity owing to the lymphodepletion conditioning regimen. There were two events of toxicities that were possibly attributed to the neoTCR transgenic T cell therapy (Extended Data Table 3). Patient 0613 developed grade 1 cytokine release syndrome, which occurred in the setting of febrile neutropaenia. Patient 1003, who had a history of treated brain metastases, developed grade 3 encephalitis, which presented as difficulty in walking, and tremulous and difficulty in writing on day 7 after infusion. Treatment of patient 1003 with high-dose corticosteroids for 4 days led to resolution of the symptoms. Eleven patients had disease progression and five patients had stable disease as their best response at their first tumour assessment (day 28 after infusion; Extended Data Table 3 and Extended Data Fig. 7f). Two of these patients had decreases in size of some of the target lesions. This included the first patient in the clinical trial, 0010 in the dose level 1 group, who had a 17% decrease in size of the sum of maximum diameter of target lesions on day 28. This patient had metastatic breast cancer and had received seven previous lines of therapy, but it could be argued that this cancer may have responded to high doses of cyclophosphamide from the preconditioning regimen, as there was limited in vivo expansion of the infused neoTCR transgenic T cells, an APOBEC signature with loss of the targeted neoantigens and HLA LOH for one of the targeted alleles detected retrospectively. Patient 1003, the last patient enrolled into the clinical trial, had metastatic non-small cell lung cancer and had received six previous lines of therapy (three at the time of informed consent and three more before neoTCR transgenic T cell infusion). This patient had stable disease on day 28 scans, with an overall sum of target lesions of −2% from baseline scans. Target lesions in the liver, lymph nodes and ovary were decreased (Extended Data Fig. 7g), but there was a concomitant increase in the size of metastatic lesions at other metastatic sites. Although no post-treatment biopsy samples were available to look for T cell infiltration or detection of the targeted neoantigens, the high percentage of neoTCR transgenic T cells detected in the periphery and the decrease in some, but not all, target lesions suggests that the therapy may have had an effect.

## Discussion

Here we developed and utilized several technologies to achieve the following: efficiently define the landscape of T cell responses to mutational neoantigens presented by over 60 HLA class I alleles; clone neoTCR genes from individual T cells circulating in peripheral blood; and genetically engineer them back into autologous T cells using non-viral gene-editing techniques. In doing so, we demonstrated that it is feasible to generate personalized T cell therapies with neoantigen-specific TCRs for ACT. When infused into patients, these neoTCR transgenic T cells circulate through the blood and were detected in tumour metastases at frequencies higher than baseline native T cells with the same neoantigen-specific TCRs.

The technology described herein also demonstrated the feasibility and safety of non-viral precision genome-engineering for the manufacture of clinical-grade gene-engineered ACT products. A similar approach has been recently used to insert a CD19 chimeric antigen receptor into the *PD1* locus[35]. Infusions of this product into patients with B cell non-Hodgkin lymphoma led to high levels of antitumour

activity[35]. The use of a HR template plasmid instead of a virus for the delivery of the inserted payload enabled rapid and personalized vector generation for both prototype testing of neoTCR candidates and the generation of plasmid material at good manufacturing practice (GMP) grade. This in turn enabled clinical-grade cell manufacture. Using this strategy, we overcame the hurdles of lengthy and expensive generation of virus-based vectors to deliver the genetic payload, which hampers their use for personalized ACT. The cloning and GMP manufacture of the HR template plasmid is rapid and cost-effective, and the precision-targeted integration in the T cell genome affords an extra level of safety compared to random integration associated with viral vectors. A further advantage of using HR plasmids is the ability to integrate payloads that exceed the packaging limits of adeno-associated viruses and other viral vectors.

The clinical trial approach described herein has several limitations. A major challenge of an approach that uses targeted personalized neoantigens is the limited ability and time to characterize each of them for protein expression and neoepitope presentation. Here we assessed gene expression and mutation truncality parameters to select neoantigens for targeting. Favouring mutations induced by oncogenic events in the first cancer clone (for example, from carcinogens such as tobacco or ultraviolet light) should result in truncal mutations, as opposed to targeting mutations induced by DNA repair syndromes or APOBEC, which have a higher likelihood for divergent heterogeneous evolution[36]. We consider targeting truncal mutations whenever possible; however, truncality determinations from a single biopsy are not perfect and additional samples, for example, in the form of ctDNA are desirable. In addition, retrospective analysis of the biopsy samples taken at the time of screening demonstrated that four of the selected neoTCRs were restricted by a HLA exhibiting LOH. This finding highlights the need for the inclusion of LOH analysis in the screening protocol before deciding which neoantigens to target[37]. Given the potential for immunoediting of highly immunogenic neoantigens, biopsy samples taken at baseline should be analysed for the status of antigen-presenting machinery molecules to rule out LOH of *HLA* alleles and alterations in *TAP* transporters or *B2M*[38–40].

Another complexity of the personalized approach results from the different affinities of the neoTCRs selected for each patient. In the current work, we initially cast a wide net based on literature that suggested that low-affinity TCRs on T cells could be beneficial in chronic viral infections[41]. To provide a benchmark for TCRs used in ACT therapies for the treatment of solid cancers, we directly compared the activity of neoTCRs used in our clinical trial with previously well-characterized TCRs to shared antigens such as MART-1, KRAS-G12D, HPV E6 and E7, mutated p53 or NY ESO-1, for which clinical data were available[21–27]. These TCRs had $EC_{50}$ activities below 10 ng ml$^{-1}$, with the exception of the HPV-E6 TCR, which had an $EC_{50}$ of 63 ng ml$^{-1}$. By comparison, only 8 out of the 37 TCRs that were selected for cell product manufacture in our neoTCR clinical trial had IFNγ $EC_{50}$ values below 10 ng ml$^{-1}$. As we learned from our own data about neoTCR T cell trafficking into the tumour and as more clinical data became available[42,43], we narrowed the TCR affinity criteria for product selection in favour of higher affinity TCRs. These criteria were applied to the last two patients (0417 and 1003) who received the treatment in the current clinical trial.

Last, the personalized neoTCR isolation, cloning, validation and selection steps resulted in a lengthy process that was heavily dependent on the quality of the tumour and PBMC samples available for analyses. The time taken to obtain additional samples and repeated analyses, particularly for screening of TCRs, delayed the ability to infuse the neoTCR cell products into patients. This could be mitigated in the future through streamlined sample acquisition and process automation. A potential solution for issues regarding TCR and neoantigen variability and the lengthy product selection process would be to use the same technology for TCR discovery and validation to generate a

pre-established library of TCRs specific for common mutations and viral antigens that would cover multiple HLA specificities.

Limited in vivo expansion of the infused neoTCR gene-edited T cells was observed in participants who received the original conditioning chemotherapy and cell products manufactured using processes v.2.0 and v.2.1 and the two low-cell doses. This in turn probably resulted in a low probability of clinical benefit. The dose-escalation study started with cell doses that may be lower than would be needed for the potential of a clinical response, especially if we consider that the total transgenic cell dose was divided by the three TCRs in many patients. In the solid tumour setting, TCR transgenic T cell clinical trials conducted by other groups have shown clinical activity in the $5-10 \times 10^9$ per TCR range, with no clear dose response beyond 10 billion cells per TCR[42,43]. The improvements in the manufacturing process during the conduct of the trial and progressing through the cell doses led to better in vivo expansion in the last patients who received treatment, getting closer to the levels that would be therapeutic in other studies[21–27].

In conclusion, we demonstrated the ability to use CRISPR-based, non-viral knockout and knock-in editing to genetically redirect T cells to mutational neoantigens in humans. This work builds on research on the use of genome editing to redirect T cell specificity with transgenic TCR engineering[44] to generate a widely applicable, tumour-specific, personalized T cell therapy for patients with solid cancers. Substitution of the endogenous TCR with a neoTCR results in T cells that only react to the mutation presented by a specific HLA, thereby providing a safe target for T cell engineering and redirection to cancer cells. With our demonstration of the feasibility and safety of this approach, neoTCR engineered T cells could be further genetically engineered to increase their functionality. The versatility of the non-viral gene-editing approach, which in a single step enables the knocking out and knocking in of several genes, predicts that future clinical approaches will be able to incorporate additional gene edits that improve T cell function and avoid T cell exhaustion. Other advances include edits that permit T cells to continue to react to antigen despite repeated antigen encounters, avoid immune suppressive factors in the solid tumour microenvironment and allow in vivo expansion without the need for lymphodepleting conditioning chemotherapy. Many potential targets have been identified from T cell biology studies in the past decades, T cell functional analyses from knockout mice, recent CRISPR screens and engineering of synthetic receptors[29,45–49]. Together, such research will provide a path to generate neoTCR engineered T cell therapies with the ability to control in vivo expansion and avoid T cell exhaustion, thereby hopefully resulting in complete and durable responses for patients with solid tumours.

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

[1]PACT Pharma, South San Francisco, CA, USA. [2]Department of Neurology and Chao Family Comprehensive Cancer Center, University of California, Irvine, CA, USA. [3]Institute for Systems Biology, Seattle, WA, USA. [4]Department of Medicine, Division of Hematology–Oncology, University of California, Los Angeles (UCLA), Los Angeles, CA, USA. [5]Division of Hematology/Oncology, Department of Medicine, University of California, San Francisco, CA, USA. [6]Jonsson Comprehensive Cancer Center at the University of California, Los Angeles, CA, USA. [7]Division of Hematology/Oncology, Department of Internal Medicine, University of California Davis Comprehensive Cancer Center, Sacramento, CA, USA. [8]Department of Medical Oncology and Therapeutics Research, City of Hope National Medical Center, Duarte, CA, USA. [9]Department of Medicine and Robert H. Lurie Cancer Center, Northwestern University, Evanston, IL, USA. [10]Clinical Research Division, Fred Hutchinson Cancer Research Center, Seattle, WA, USA. [11]Thoracic Oncology Service, Division of Solid Tumor Oncology, Department of Medicine, Memorial Sloan Kettering Cancer Center, Weill Cornell Medical College, New York, NY, USA. [12]Division of Biology and Biological Engineering, California Institute of Technology, Pasadena, CA, USA. [13]These authors contributed equally: Susan P. Foy, Kyle Jacoby, Daniela A. Bota. [14]These authors jointly supervised this work: Antoni Ribas, Arati V. Rao, Stefanie J. Mandl. ✉e-mail: sfoy@pactpharma.com; ARibas@mednet.ucla.edu; smandl@pactpharma.com

## Methods

### Clinical trial design

This phase Ia trial was a multicentre 3 + 3 dose-escalation study. The primary objectives were to evaluate safety and tolerability, determine a maximum tolerated dose and evaluate manufacturing feasibility (NCT03970382). Safety and feasibility were assessed for the infusion of up to three distinct neoTCR T cell products for each patient. Patients were treated with neoTCR$^+$ T cells at doses of $4 \times 10^8$ (dose level 1, $1.3 \times 10^8$–$4 \times 10^8$ cells per TCR), $1.3 \times 10^9$ (dose level 2, $4 \times 10^8$–$1.3 \times 10^9$ cells per TCR) or $4 \times 10^9$ (dose level 3, $1.3 \times 10^9$–$4 \times 10^9$ cells per TCR). The total number of gene-edited cells at a given dose level remained the same regardless of the number of neoTCR T cell products infused (level 1, 2 or 3). Only participants who received three neoTCR products contributed to clearing a dose level. The first four patients received a conditioning regimen of 30 mg m$^{-2}$ fludarabine and 300 mg m$^{-2}$ cyclophosphamide intravenously (days −5 to −3; 3 doses each) and it was modified to 30 mg m$^{-2}$ fludarabine (intravenously; days −6 to −3; 4 doses) and 600 mg m$^{-2}$ cyclophosphamide (intravenously; days −5 to−3; 3 doses) for subsequent patients. neoTCR T cells were infused on day 0, with consecutive infusion of up to three neoTCR T cell products. After safety was evaluated at each dose level, additional participants were eligible for expansion at the cleared dose level along with 500,000 IU m$^{-2}$ of low-dose IL-2 (aldesleukin) administered subcutaneously twice a day from days 1–7 starting on day 1.

### Study oversight

The protocol was approved by the institutional review board at each of the following sites enrolling patients: City of Hope, Duarte, CA; University of California Los Angeles, Los Angeles, CA; University of California, Irvine Medical Center, Orange, CA; University of California, Davis, Sacramento, CA; University of California, San Francisco, San Francisco, CA; Northwestern University Medical Center, Chicago, IL; Memorial Sloan Kettering Cancer Center, New York, NY; Tennessee Oncology, Nashville, TN; and Fred Hutchinson Cancer Research Center, Seattle, WA. The trial was conducted in accordance with the principles of the Declaration of Helsinki. An independent data and safety monitoring committee regularly reviewed safety data. All patients provided informed written consent. The trial protocol and statistical analysis plan were designed in a collaboration between the sponsor (PACT Pharma) and the authors.

### Patients

Patients were eligible for screening to identify TCRs if they met the following criteria: were ≥18 years of age; had one of the following metastatic solid tumour types: urothelial carcinoma, melanoma, non-small cell lung carcinoma, head and neck squamous cell carcinoma, colorectal cancer, ovarian cancer, hormone-receptor positive breast cancer, triple-negative breast cancer, or prostate cancer; had disease progression after at least one available standard therapy with no additional curative options available; and measurable disease per response evaluation criteria in solid tumours (RECIST) v.1.1. Inclusion criteria for TCR identification included providing a tumour biopsy (fresh or archival tumour biopsy within 1 year of consent) for sequencing and neoantigen prediction, PBMCs for T cell isolation and a willingness and ability to undergo leukapheresis for cell product manufacture. To decrease the risk of decline during TCR product selection and cell therapy manufacture and infusion, patients were required to have a life expectancy of >6 months, an Eastern Cooperative Oncology Group status of 0 or 1 and adequate haematological and organ function.

Between consent and neoTCR product selection, patients were able to be treated with anticancer therapies. Patients were required to discontinue therapy within 2 weeks or five half-lives (whichever was shorter) before leukapheresis. Following successful neoTCR product selection and before leukapheresis, patients were required to meet the initial screening criteria with adequate haematological and organ function. Patients with asymptomatic brain metastasis were included. Eligible patients had an absolute lymphocyte count of at least 500 cells per cubic millimetre. During cell product manufacture, patients could receive bridging therapy at the investigator's discretion, after which repeat baseline imaging was performed. Before receiving conditioning chemotherapy and cell infusion, patients were required to have no significant changes in status compared with the initial eligibility criteria.

### Safety and response assessments

Incidence and nature of adverse events to define dose-limiting toxicities was documented using the National Cancer Institute's Common Terminology Criteria for Adverse Events v.5.0 except for cytokine release syndrome and neurotoxicity, which were evaluated per the American Society for Transplantation and Cellular Therapy consensus criteria. The dose-limiting toxicity assessment window was from day 0 to day 28. A safety review team was chartered to review safety after each dose level and before opening up enrolment for IL-2 treatment. Response assessment for overall response rate was determined by investigator assessment using RECIST v.1.1 on day 28 and confirmed by repeat assessment ≥4 weeks after initial documentation.

### Biospecimen collection and processing

Archival or fresh tumour biopsy specimens were formalin-fixed and paraffin-embedded (FFPE). Paraffin embedding was performed at the study site or at CellCarta. Tumour sectioning was performed by CellCarta, and cut sections were used for sequencing at Personalis or for fluorescence microscopy. Screening peripheral blood samples for sequencing were collected in K2EDTA tubes, processed at Precision for Medicine and sequenced at Personalis. Peripheral blood samples for T cell identification at screening or on-treatment analysis were collected in ACD or CPT tubes and shipped to Precision for Medicine for PBMC isolation and cryopreservation. For clinical cell product manufacture, a whole leukapheresis product was obtained from the patient at the study site and shipped overnight to the study sponsor. Samples from each infusion cell product and final clinical cell manufacture product were cryopreserved. Serum and plasma were collected and cryopreserved until analyses.

### Neoantigen prediction and truncality estimation

Tumour biopsy samples from patients and the matched normal sample from PBMCs were sequenced to identify expressed NSMs. Although most of the analysis was the same, there were 3 versions of the pipeline used throughout the clinical trial and the differences are described below. Samples from patients 0010, 0026, 0503, 0506, 0603, 0604, 0605, 0612 and 0613 were processed using v.1.0. Samples from patients 0030, 0038, 0404, 0411 and 0611 were processed using v.2.0. Samples from patients 0417 and 1003 were processed using v.30. In brief, WES and RNA-seq from a recent tumour sample in a FFPE section and PBMCs were performed using an Illumina HiSeq 2500 platform or a NovaSeq 6000 platform (Illumina). First, WES sequences were aligned to the human reference genome build 37 (GRCh37/hg19) using BWA-MEM[50]. NSMs identified by at least two mutation callers among VarScan2 and MuTect or MuTect2, VarDictJava and Strelka2 were retained as potential neoantigens[51–54]. RNA-seq sequences were mapped to the human genome, quantified and normalized using STAR and RSEM[55]. A minimum of 1 RNA-seq read was used for conformation of expression; however, transcripts per million and expression values were further considered at the time of product selection. Next, the neoantigen sequences and the patient's HLA types identified from the patient's PBMC WES using OptiType[56] were used as input for HLA−peptide binding affinity prediction with netMHCpan. In v.3 of the software, OptiType alignment with RNA-seq reads to individual restricting HLAs to access expression levels were routinely reported. For v.1, netMHCpan 3.0 was used, whereas v.2 used netMHCpan 4.0 and v.3 used

netMHCpan 4.1 (ref. [57]). In addition, the truncality and cancer cellular fraction of the detected NSMs were predicted on the basis of the WES results and were available at the time of product selection, with preference given to truncal NSMs. In brief, the read alignment files from WES of tumour biopsy samples and matched PBMCs generated previously were used as input for copy number segmentation, ploidy and tumour purity estimation, and allelic copy number profiling using Sequenza[58]. This step was followed by PyClone analysis as previously described for NSM truncality estimation[59]. Finally, the HLA–peptide complexes with predicted binding affinities among the top 2% ranking with respect to each HLA were selected, and only the peptides with confirmed expression by RNA-seq were included. A maximum of 352 selections were made per patient. In v.3 of the software, the first five epitopes chosen consisted of recurrent driver mutations[60] regardless of ranking if they were present and expressed. Prioritized HLA–peptide complexes were used to generate protein reagents. Neoepitopes that were derived from recurrent driver mutations were noted at the time of product selection.

### Retrospective sequence analysis after dosing and screening samples

Amplicon sequencing-based TCR assays were conducted at Personalis to further interrogate the immune repertoire in tumour biopsy samples and the CDR3 reads. The frequency from TCRα and TCRβ were reported to evaluate the neoTCR trafficking after dosing[61]. To further ensure the integration of the neoTCR and its expression in the biopsy samples taken after infusion, the tumour sequencing reads from WES and RNA-seq were aligned to the gene-editing payload containing neoTCR TCRβ, TCRα, P2A and partial *TRAC* chain following similar protocols described in the main text. The tumour sequencing reads from RNA-seq were used as input to the MiXCR program[62]. The TCRα and TCRβ CDR3 sequences in the tumour biopsy samples were identified and compared with neoTCR CDR3 sequences. Five distinct base changes in a short stretch of a codon-optimized constant region served as an effective barcode for identifying specific neoTCRs. The sample somatic mutational signatures for patients was determined using MuSiCa from screening WES data[63]. This program compares somatic signature profiles to previously published COSMIC somatic signatures (https://cancer.sanger.ac.uk/signatures/)[64].

HLA LOH was analysed using both WES and RNA-seq data[65] using two complimentary approaches. First, WES data were used to access allelic copy number at HLA loci using Sequenza[58], with a copy number state of zero denoting putative LOH. To determine which HLA allele was lost, single nucleotide polymorphism (SNP) frequencies between alleles were compared with a binomial generalized linear model, and the allele with the lower SNP frequency was assigned to the lower copy number state. Unique SNPs for HLA alleles were derived from IMGT (v.3.40.3)[66]. Second, the allelic imbalance at 26 HLA loci was calculated in DNA and RNA, for which the allelic imbalance is the normalized ratio of sequencing depth at SNP positions between alleles at a locus. A higher allelic imbalance in DNA corresponds to a greater difference in copy number between alleles, whereas a higher allelic imbalance in RNA represent a greater difference in the expression of one allele relative to the other. Samples with LOH are expected to have higher allelic imbalance in DNA and RNA because LOH reduces the sequencing depth at SNP positions for the lost allele. Allelic copy number and allelic imbalance in DNA and RNA were then manually reviewed to confirm the HLA LOH classification.

Somatic variants from WES data were used for a custom ctDNA assay made by Natera as previously described[67]. In brief, this assay designs multiplex PCR primers for 16 truncal variants and up to 16 custom variants of interest. Plasma from longitudinal samples had ctDNA isolated and amplified and subsequently sequenced on an Illumina sequencer. All ctDNA time points were normalized by quantifying the mean tumour molecules per ml of plasma.

### Peptide–HLA protein synthesis

Libraries of peptide–HLA complexes were generated by assembling single-chain trimers[68], in which neoantigen peptides were fused in sequence to B2M domains and the HLA, each domain being linked with $(G4S)_4$ motifs. In brief, double-stranded oligonucleotides (from Integrated DNA Technologies) were ligated into pcDNA vectors encoding linkers, B2M and the corresponding HLA, and the expression sequence amplified by PCR. Linear amplicons were transfected into Expi293F cells (Thermo Fisher) following the manufacturer's recommendations. Cells were collected after 5 days, and supernatants were clarified using 96-well Pall filters. Proteins were biotinylated with biotin ligase BirA (Avidity) for 2 h at room temperature. Biotinylated proteins were purified using IMAC (Ni-sepharose) with the Phynexus Phytip system, and buffer-exchanged into HBS using Thermo Fisher Zeba Spin 7k MWCO plates or by $Zn^+$-loaded HiTrap capto-chelate resin in tandem with HiTrap desalting columns to remove imidazole. The protein concentration was determined by absorbance at 280 nm.

### neoTCR isolation

Each purified patient peptide–HLA protein was multimerized, fluorescently labelled and DNA barcoded. The barcode followed the structure adaptor-UMI-barcode-UMI-adaptor (Integrated DNA Technologies). In brief, biotinylated peptide–HLA proteins and biotinylated DNA barcodes were mixed at a 3:1 molar ratio and then complexed into fluorescently labelled multimers with PE-streptavidin or APC-streptavidin (Life Technologies) at 4:1 molar ratio of streptavidin to biotin. Peptide–HLA multimers for each patient were then pooled, concentrated and used to stain cells.

neoTCRs were isolated from a patient's own peripheral blood. In brief, PBMCs were enriched for CD8 T cells by negative selection (Miltenyi Biotec) with the addition of CD16 and CD56 markers (R&D Systems) to prevent the loss of activated CD8 T cells. Enriched CD8 T cells were then stained with the pooled library of neoantigen peptide–HLA multimers plus a panel of fluorescently labelled antibodies against cell surface markers, including CD39, CD103 and CD95 (see Supplementary Table 4 for a list of the reagents, and Supplementary Fig. 1 for the gating strategy). Next, antigen-experienced CD95$^+$, multimer$^+$ CD8$^+$ T cells expressing neoTCRs were single-cell sorted using a FACSAria III (BD Biosciences). DNA barcodes from each sorted cell were sequenced and used to identify the neoantigen peptide–HLA binding target of the TCR. TCRα and TCRβ chains were amplified by PCR with reverse transcription and sequenced (MiniSeq or MiSeq, Illumina). TCRα and TCRβ reads were used to identify V and J chains and to reconstruct CDR3 sequences as well as full-length VDJ regions by leveraging the IMGT library[69] with the MiXCR program[62].

### Homology directed repair template generation

Paired TCRα and TCRβ variable regions from neoepitope-specific T cells were amplified by PCR using the corresponding variable region-specific primers. The purified PCR products were assembled with constant regions and homology arms to generate patient-specific HR template plasmids. The patient-specific HR template plasmid was designed to direct the integration of the gene cassette into the first exon of *TRAC*. The payload consisted of the following structure: P2A, HGH signal sequence, neoTCR β-chain, furin cleavage site, P2A, human growth hormone (HGH) signal sequence, neoTCR α-variable region and partial TRAC constant chain. Homology arm sequences homologous to the *TRAC* locus flanked the plasmid payload and were 1,000 base pairs each. The templates were verified by Sanger sequencing and agarose gel electrophoresis.

### neoTCR T cell generation

CD4 and CD8 T cells were positively enriched from healthy donor leukapheresis products by magnetic selection (Miltenyi) and activated

for 48 h with CD3 and CD28 stimulation (TransACT, 1:17.5 by volume; Miltenyi) in T cell medium (TexMACS medium, Miltenyi, supplemented with 12.5 ng ml$^{-1}$ IL-7 and IL-15, Miltenyi, and 3% human AB serum, Valley Biomedical). After activation, the T cells were centrifuged and resuspended in P3 buffer (Lonza). CRISPR–Cas ribonucleoproteins (RNPs) were formulated by complexing guide RNAs targeting *TRAC* and *TRBC* (Synthego) to spCas9 protein (Aldevron) in a 6:1 molar ratio. The patient-specific HR template and RNPs were mixed with the cell suspension, electroporated (Lonza, X-unit, EO-115) and transferred into T cell medium in a 24-well G-rex (Wilson Wolf) for 4–5 days with changes of the medium as needed.

### neoTCR binding and affinity analyses

Specific binding of the patient-specific neoTCR was confirmed by flow cytometry. Biotinylated peptide–HLA molecules were fluorescently labelled with PE-streptavidin (Thermo Fisher) and biotin-labelled dextran (500 kDa, Nanocs) to generate a dextramer for staining as previously described[70]. Cells were stained with a fixable viability dye, the matched peptide–HLA dextramer to measure neoTCR binding and CD4 and CD8 (see Supplementary Table 4 for reagents). Cells were permeabilized and stained intracellularly with 2A antibody to assess gene editing. Gene editing was confirmed if ≥5% of the T cells stained positive for 2A. TCR identity was confirmed if ≥5% of neoTCR$^+$ T cells stained positive for the matched peptide–HLA dextramer. CD8-independent binding was confirmed when 2A$^+$neoTCR$^+$ binding on CD4$^+$ T cells was ≥50% of the 2A$^+$neoTCR$^+$CD8$^+$ T cells. Otherwise, edited CD4 T cells were considered to have weak or no binding (CD8-dependent). An affinity score was generated as a metric to further quantify CD8-dependent or CD8-independent binding using the following formula: Affinity score = (CD4$^+$neoTCR$^+$2A$^+$/CD4$^+$2A$^+$) + (CD8$^+$neoTCR$^+$2A$^+$/CD8$^+$2A$^+$). In general, this results in the following TCR calling: <0.25 = non-binders, 0.25–1.25 = CD8 dependent, and >1.25 = CD8 independent.

### Plate-based antigen stimulation of neoTCR cells

Streptavidin-coated plates (Eagle Biosciences) were pre-incubated with cognate or control peptide–HLA molecules at various concentrations for 2–5 h at room temperature or 16–30 h overnight at 4 °C. T cells electroporated with neoTCRs or control TCRs were then stimulated on the plates in T cell culture medium (TexMACS supplemented with 3% human AB serum and 1% penicillin–streptomycin, Gibco) at 37 °C and 5% $CO_2$.

### IFNγ secretion and product selection

The supernatant from the plate-based stimulation assay (overnight using 0.1–1,000 ng ml$^{-1}$ peptide–HLA) was collected and analysed by CBA (Human Th1/Th2 Cytokine kit II, BD Biosciences), acquired on an Attune NxT flow cytometer (Thermo Fisher) and EC$_{50}$ values were calculated for IFNγ using GraphPad Prism. Product selection was performed considering target truncality, expression, TCR functionality and neoantigen and HLA diversification.

### Cell killing in a colorectal cancer cell line

A tumour biopsy and PBMCs were obtained from a treatment-naive patient with colorectal adenocarcinoma (Asterand). A neoTCR was isolated from the PBMC sample targeting a mutation in *COX6C* (R20Q peptide amino acids 18–46, HLA-A:02:01). The SW620 colorectal cancer cell line (CCL-227, American Type Culture Collection, expanded to a cell bank and tested negative for mycoplasma) was transduced with IncuCyte NucLight Red lentivirus (Sartorius) and sorted for high dye (red) expression. The SW620 cells were then transfected with guide RNA and Cas9 RNPs and single-stranded homology directed repair template containing the desired neo-antigen point mutation (R20Q) and PAM-ablating mutation in *cis* under the control of the endogenous regulatory elements. Genotyping confirmed editing and clonal cell populations were isolated by limiting dilution cloning and single-cell

sorting. Wild-type or mutant SW620 cells, expressing endogenous levels of HLA and neoepitope, were then incubated with neoTCR-specific T cells in an IncuCyte (Sartorius) system for 24 h to determine target cell killing.

### Clinical manufacturing

**Plasmid manufacture.** All clinical products were manufactured in the PACT Plasmid and Cell GMP manufacturing facilities (South San Francisco, CA) following clinical manufacturing protocols. neoTCR-P1 plasmid was propagated from patient-specific HR template plasmid generated for TCR validation, using selected patient-specific HR template plasmid as source material for GMP plasmid manufacturing. Patient-specific plasmid reagent was transformed into *Escherichia coli* 5α (Aldevron) competent cells. Transfected cells were plated, and a single isolated colony was used for seed culture growth then transferred to inoculate the main fermenter. At the end of fermentation, the culture was collected and centrifuged to collect the cell paste. The cell paste was lysed, RNA enzymes digested, then clarified by flocculation and depth filtration. During purification, the clarified lysate was processed using anionic exchange chromatography then hydrophobic interaction chromatography. Purified plasmid was concentrated and diafiltered into formulation buffer by hollow-fibre cartridge tangential flow filtration, sterile filtered and frozen until cell manufacture.

**Clinical cell manufacture.** Patient leukapheresis products were received from clinical sites after overnight shipment. CD4 and CD8 T cells were positively selected using ClinicMACs Prodigy (Miltenyi). Up to 715 × 10$^6$ cells were seeded in culture medium (TexMACS medium with 3% human AB Serum, V2.0; or PRIME-XV medium (Irvine Scientific), V2.1 or V3.0; Supplementary Table 3) supplemented with cytokines (IL-7 and IL-15, each at 12.5 ng ml$^{-1}$) and activated for 44 h with TransACT (1:17.5 by volume) in a G-Rex 100M CS (Wilson Wolf). Cells were then collected and electroporated using a Lonza Nucleofector (LV-unit, EO-115, V2.0 or V2.1) or a pre-commercial version of the CTS Xenon Electroporation system (Thermo Fisher, V3.0) with RNPs and patient-specific plasmid DNA to express a patient-specific neoTCR. Each individual lot was then expanded in a G-Rex in cytokine-supplemented growth medium (TexMACS medium with 3% human AB Serum, V2.0; or PRIME-XV medium with 2% Physiologix serum replacement (Nucleus Biologics), V2.1, V3.0), with medium exchanges and splits to additional G-Rex vessels as appropriate. On day 13, cells were collected and cryopreserved in a 50:50 mix of Plasma-Lyte-A (Baxter) plus 2% human serum albumin (Grifols) and CS10 (Stemcell Technologies).

### neoTCR staining and T cell functional assays

**IFNγ secretion at clinical lot release.** An aliquot of fresh cells was taken on the day of collection and tested using the plate-based stimulation assay described above (24 h stimulation). Controls consisted of 'mismatch' peptide–HLA-coated plates. The supernatant was collected and analysed using IFNγ ELISA kits (Quantikine ELISA Human IFNγ Immunoassay kit, R&D Systems) or using an ELLA Simple Plex Immunoassay platform (Simple Plex for Human IFNγ, Bio-techne). EC$_{50}$ values were calculated using GraphPad Prism.

**Proliferation assay.** An aliquot of the cryopreserved final cell product was thawed, washed and rested for 3 days in T cell recovery medium (TexMACS with 3% human AB serum and 12.5 ng ml$^{-1}$ IL-7 and IL-15). Rested cells were labelled with ViaFluor (Biotium) and incubated for 10 min at 37 °C, followed by a 30-min incubation with stop solution (day 0). ViaFluor-stained cells were then used for the plate-based stimulation assay (0.1–1,000 mg ml$^{-1}$ peptide–HLA) for 22–26 h, supplementing the medium with 5 ng ml$^{-1}$ IL-7 and IL-15. No stimulation, mis-matched peptide–HLA (1,000 ng ml$^{-1}$) and TransACT (1:17.5) were used as controls. The next day (day 1), TransACT samples were washed twice and all samples were removed from stimulation and transferred

to a fresh plate in T cell culture medium and placed at 37 °C and 5% $CO_2$ for 72 h (day 4). ViaFluor cells were collected on day 0 (before stimulation) or day 4 (after stimulation) and stained for flow cytometry analysis (see Supplementary Table 4 for reagents and Supplementary Fig. 2 for the gating strategy). Cells were labelled with a fixable viability dye and stained with neoTCR-matched dextramer to measure neoTCR surface expression, CD8, CD4 and TCRαβ. Cells were then fixed, permeabilized and stained intracellularly with 2A antibody to assess total gene editing. Cells were fixed and acquired on an Attune NxT flow cytometer. Proliferation $EC_{50}$ is defined as the neoantigen peptide–HLA concentration at which ViaFluor mean fluorescence intensity reaches half of the minimum ViaFluor mean fluorescence intensity when the dose–response curve is fitted with a sigmoidal trend.

**Intracellular cytokine staining.** An aliquot of thawed and rested cells from the proliferation assay were stimulated for 16 h using the plate-based stimulation assay (0.1–1,000 mg ml$^{-1}$ peptide–HLA, 100,000 cells per well). No peptide–HLA was used as a control. T cell culture medium was supplemented with CD107a antibody, brefeldin A and monensin protein secretion inhibitors. The next day, cells were stained with a fixable viability dye and CD4 and CD8 surface markers, fixed and permeabilized and stained intracellularly for IFNγ, TNF and IL-2 (see Supplementary Table 4 for reagents and Supplementary Fig. 3 for the gating strategy). Cells were fixed and acquired using an Attune NxT flow cytometer.

## Flow cytometry analysis of neoTCR cells in manufactured products and peripheral blood

Cryopreserved T cell products or PBMC specimens were thawed, washed and labelled with fixable viability dye. For identification of cells expressing the neoTCR, cells were incubated with a multimer reagent prepared using cognate peptide–HLA molecules[70], then stained with a panel of surface antibodies for pharmacokinetic analysis (see Supplementary Fig. 4 for the gating strategy). Transgene expression was further identified by intracellular staining with 2A peptide antibody. Additional staining for phenotypic markers was performed on thawed manufactured products or PBMC specimens (see Supplementary Table 4 for a list of all flow cytometry reagents, and Supplementary Figs. 5–8 for the gating strategies). PBMC specimens taken after treatment were first enriched for neoTCR$^+$ cells after peptide–HLA multimer staining using anti-APC magnetic enrichment beads (Miltenyi). neoTCR$^+$ counts per μl were calculated using the following formula: absolute lymphocyte count × (CD5$^+$ (%) of live lymphocytes) × (neoTCR$^+$ (%) of CD5$^+$). Matching absolute lymphocyte count data were not available for all time points. Data were acquired using an Attune NxT cytometer and analysis was performed using FlowJo (BD Biosciences) or FCS Express (De Novo Software).

## Serum cytokine analysis

Serum protein concentrations were measured by Precision for Medicine using an electrochemiluminescence immunoassay (Meso Scale Discovery). V-Plex Proinflammatory panel 1 was used for the following cytokines: IFNγ, IL-2, IL-6, IL-8, IL-10, IL-12p70, IL-13 and TNF. V-Plex Cytokine panel 1 was used for the following cytokines: GM-CSF, IL-7 and IL-15. Single V-Plex and U-plex assays were used for IL-1RA and IL-2Rα, respectively. Analysis was performed using a MESO QuickPlex SQ 120 instrument and Discovery Workbench 4.0 software (Meso Scale Discovery).

## Fluorescence microscopy of tumour FFPE sections

Tumour FFPE sections were imaged using RNAscope fluorescence ISH and immunofluorescence. RNAscope combined with immunofluorescence was performed using a RNAscope Multiplex Fluorescent Detection kit v2 (323110, Advanced Cell Diagnostics) combined with a RNA-Protein Co-detection Ancillary kit (323180, Advanced Cell

Diagnostics). The protocol was adapted from the vendor's documentation entitled 'RNAscope Multiplex Fluorescent v2 Assay combined with Immunofluorescence–Integrated Co-Detection Workflow (ICW)' (MK 51–150/Rev A/ effective date 10 May 2020). The ISH component of the assay uses the instructions in chapter 4 of the RNAscope Multiplex Fluorescent Reagent kit v2 user manual (323100-USM). Vendor instructions were followed except for the following modifications: (1) Tris buffer saline with Tween-20 (1×) (Cell Signaling Technology) was used instead of PBS with 0.1% Tween-20 (1×); (2) 4% formaldehyde in PBS (formed from Pierce 16% formaldehyde) was used instead of 10% neutral buffered formalin; (3) Fluoromount-G (SouthernBiotech) was used instead of Prolong Gold Antifade mountant; and (4) CitriSolv was used instead of xylene. Sections were stained with anti-CD3 (clone EP4426, Abcam; anti-rabbit AF647, Thermo Fisher), Vector2A RNAscope probe to identify neoTCR-edited cells (Advanced Cell Diagnostics; Opal 570, Akoya Biosciences) and DAPI (Advanced Cell Diagnostics).

## Statistical analysis

No formal hypothesis was tested in the phase I study. Design considerations were not made with regard to explicit power and type I error considerations, but were made to obtain preliminary safety, feasibility, pharmacokinetics, pharmacodynamics and antitumour activity information in this population. Dose escalation was conducted in a traditional 3 + 3 design, and each dose level was cleared with 3 patients treated with 3 TCRs. Measurements were taken from distinct samples. neoTCR editing and IFNγ production at product release are reported as the average of two replicate tests from the same sample. For correlations, data were first tested for normal or log-normal distributions. Pearson's correlations were performed on normally distributed data, otherwise Spearman's correlation was performed. Additional statistical tests are indicated in each figure legend.

## Reporting summary

Further information on research design is available in the Nature Portfolio Reporting Summary linked to this article.

## Data availability

The following publicly available datasets were utilized: ExAc (3.1, https://gnomad.broadinstitute.org/downloads#exac-variants), dbSNP (v146, ftp://ftp.broadinstitute.org/bundle), GATK Resource Bundle (hg19/Grch37, ftp://ftp.broadinstitute.org/bundle), Human Proteome (Homo_sapiens.GRCh37.75.pep.all.fa, http://ensembl.org/), IMGT (TCR/HLA, 3.1.17, http://www.imgt.org/), RefSeq (1052019, ftp://hgdownload.cse.ucsc.edu/goldenPath), TCGA (v.1.0, https://portal.gdc.cancer.gov/) and Broad Institute (hg19, ftp://ftp.broadinstitute.org/bundle). The TCR sequences from the current study are provided in the supplementary files, and the genomics data are available following reasonable request from the European Genome-Phenome Archive repository.

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

**Acknowledgements** We thank the patients and their families and caregivers who participated in this study; and present and previous employees of PACT Pharma who contributed to this work, including the Upstream Operations, Process Development, Plasmid Manufacturing, Cell Manufacturing, Clinical Development, Clinical Operations, Quality Control, Quality Assurance and Program Management teams. The work was funded by PACT Pharma. J.R.H. and A.R. are funded by the Parker Institute for Cancer Immunotherapy. A.R. is funded by the National Institutes of Health (grant R35 AI197633). D.Y.O. is supported by a Young Investigator Award from the Prostate Cancer Foundation and a Damon Runyon Clinical Investigator Award (CI 110-21). K.M.C. is supported by a UCLA Tumor Immunology Training Grant (NIH T32CA009120), the Cancer Research Institute Irvington Postdoctoral Fellowship Program and a V Family Foundation Gil Nickel Melanoma Research Fellowship.

**Author contributions** S.P.F., K.J., T.H., Z.P., E.S., C.L.W., W.L., A.V.R., S.J.M. and A.R. wrote the first draft of the manuscript. A.V.R., R.F., A.S. and T.S.-S. designed, initiated and ran the clinical study protocol. A.V.R., A.S., D.A.B., D.Y.O., B.C., M.A., Y.Y., A.J.S., J.A.S. and S.M.L. enrolled, treated and cared for patients on the clinical study protocol. Z.P., E.S., Y.M., K.M.C. and C.S. performed and interpreted bioinformatics analysis. M.T.B., O.D., K.H. and M.C.Y. developed or conducted work related to peptide–HLA protein synthesis. C.L.W., S.P., M.T.B., D. An., B.Y. and B.B.Q. created the process, acquired and/or analysed data for neoTCR T cell isolation. K.J., W.L. and R.M. developed the method or performed and analysed the studies for non-viral precision TCR replacement. A.J.L. developed and conducted the functional cytotoxicity experiments. B.S., A.C. and J.M. developed the methods, performed studies and analysed or interpreted data for product selection. Cell and plasmid manufacturing processes were created and supervised by I.M., D. Anaya. and L.D. Flow cytometry was performed, analysed or supervised by S.P.F., T.H., S.G., E.Y.-H.H., A.H., M.K., W.W., L.S., E.H., V.M. and B.P. (for the final cell product) and S.P.F., T.H., S.G., E.H. and A.H. (after infusion). T.H. analysed serum cytokines. A.H.C.N. and Y.L. performed fluorescence microscopy and together with J.R.H. scored and interpreted the staining results. S.P., M.T.B., A.F., S.J.M., D.B., J.R.H. and A.R. conceived the initial idea and provided scientific input for neoTCR cell isolation for personalized adoptive cell therapy. All authors proofread and approved the final manuscript.

**Competing interests** S.P.F., K.J., T.H., Z.P., E.S., Y.M., W.L., S.P., C.L.W., B.Y., O.D., K.H., B.S., A.C., M.T.B., I.M., W.W., M.K., S.G., E.H., A.H., D. An, A.J.L., B.B.Q., C.S., D.Anaya, L.S., E.Y.-H.H., V.M., J.M., L.D., B.P., R.M., M.C.Y., R.F., A.S., T.S.-S., A.F., A.V.R. and S.J.M. were employees of PACT Pharma during the conduct of this work. S.P., M.T.B., D.B., J.R.H. and A.R. are scientific co-founders of PACT Pharma. K.M.C. has received consulting fees from PACT Pharma and Tango Therapeutics and is a shareholder in Geneoscopy. D.Y.O. has received research support from Merck, PACT Pharma, the Parker Institute for Cancer Immunotherapy, Poseida Therapeutics, TCR2 Therapeutics, Roche/Genentech and Nutcracker Therapeutics. B.C. has advisory roles with the following companies: Iovance Biotherapeutics, IDEAYA Biosciences, Sanofi, OncoSec, Nektar, Genentech and Instil Bio. D.Y.O. has also received research funding from the following companies: Iovance Biotherapeutics, Bristol-Myers Squibb, Macrogenics, Daiichi Sankyo, Merck, Karyopharm Therapeutics, Infinity Pharmaceuticals, Advenchen Laboratories, Idera, Neon Therapeutics, Xencor, Compugen, PACT Pharma, RAPT Therapeutics, Immunocore, Lilly, IDEAYA Biosciences, Tolero Pharmaceuticals, Ascentage Pharma, Novartis, Atreca, Replimune, Instil Bio, Trisalus and Kinnate. J.A.S. reports honorarium from Iovance, Apixogen, Jazz Pharm, and research with BMS, PACT Pharma and Corvus. A.J.S. reports consulting or advising roles to J&J, KSQ Therapeutics, BMS, Enara Bio, Perceptive Advisors, Heat Biologics and Iovance Biotherapeutics. Research funding: GSK (Inst), PACT pharma (Inst), Iovance Biotherapeutics (Inst), Achilles therapeutics (Inst), Merck (Inst), BMS (Inst), Harpoon Therapeutics (Inst). A.R. has received honoraria from consulting with Amgen, CStone, Merck, and Vedanta, is or has been a member of the scientific advisory board and holds stock in Advaxis, Appia, Apricity, Arcus, Compugen, CytomX, Highlight, ImaginAb, ImmPact, ImmuneSensor, Inspirna, Isoplexis, Kite-Gilead, Lutris, MapKure, Merus, PACT Pharma, Pluto, RAPT, Synthekine and Tango, has received research funding from Agilent and from Bristol-Myers Squibb through Stand Up to Cancer (SU2C), and patent royalties from Arsenal Bio. D.A.B., A.H.C.N., Y.L., M.A., Y.Y. and S.M.L. report no conflicts of interest for this work.

**Additional information**
**Correspondence and requests for materials** should be addressed to Susan P. Foy, Antoni Ribas or Stefanie J. Mandl.

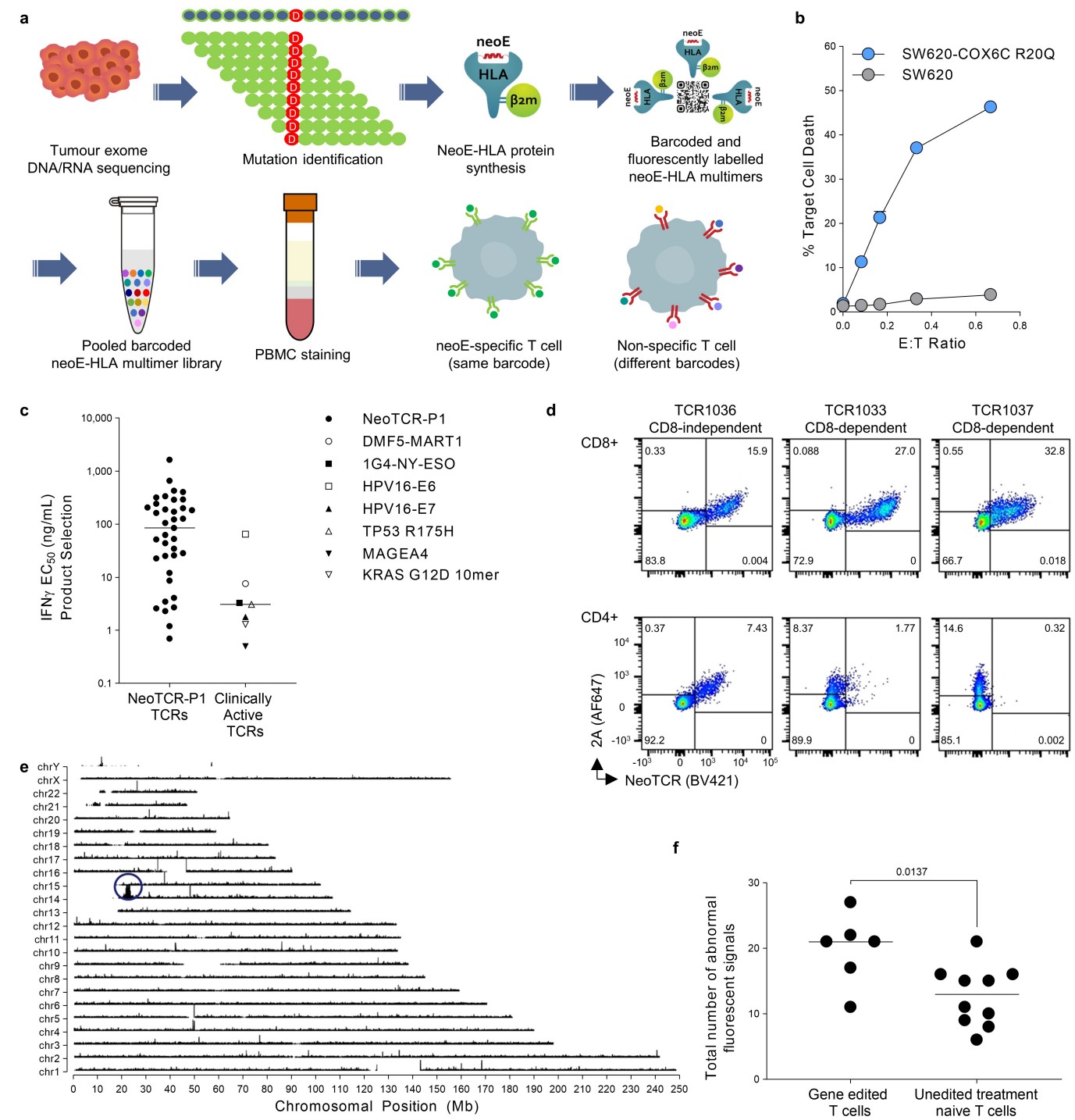

**Extended Data Fig. 1 | NeoTCR isolation, cytotoxicity, potency, gene editing and gene insertion. a**) Neoantigen-specific T cell capture. **b**) NeoTCR specific killing of an SW620 COX6C-R20Q mutant colorectal cancer cell line. Healthy donor T cells engineered to express a neoTCR from the blood of a patient with colorectal cancer targeting the COX6C-R20Q mutation, cocultured with either the parental SW620 cell line (without R20Q mutation), or with SW620-COX6C R20Q. **c**) Potency (IFNγ EC$_{50}$) of neoTCRs isolated from the 16 patients compared with seven clinically active TCRs. **d**) Example gene editing (as measured by staining for 2A peptide) and neoTCR binding on CD4 and CD8 T cells for the three TCRs in a manufactured cell product. TCR1036 showed 2A expression and neoTCR binding of dextramer in CD4 and CD8 T cells, considered CD8-independent. TCR1033 and TCR1037 showed only 2A expression but no

neoTCR binding by dextramer when transfected into CD4 T cells, considered CD8-dependent. **e**) Targeted locus amplification (TLA) was performed on 0010 TCR445 drug product. Primers specific for transgene and integrated transgene were used to amplify TLA processed genomic DNA. High coverage at the chromosome 14 integration site was observed (blue circle), indicating on-target *TRAC* transgene integration. A similar peak was not observed at chromosome 7, the site of TRBC knockout. **f**) Six clinical drug products from three patients were analysed using fluorescent *in-situ* hybridization (FISH) for chromosomal anomalies involving chromosome 7 and chromosome 14. All abnormal signals from each drug product tested were summed and compared to the total number of abnormal signals found in unedited cells from 10 separate donors. A p value was generated using an unpaired two-tailed t-test.

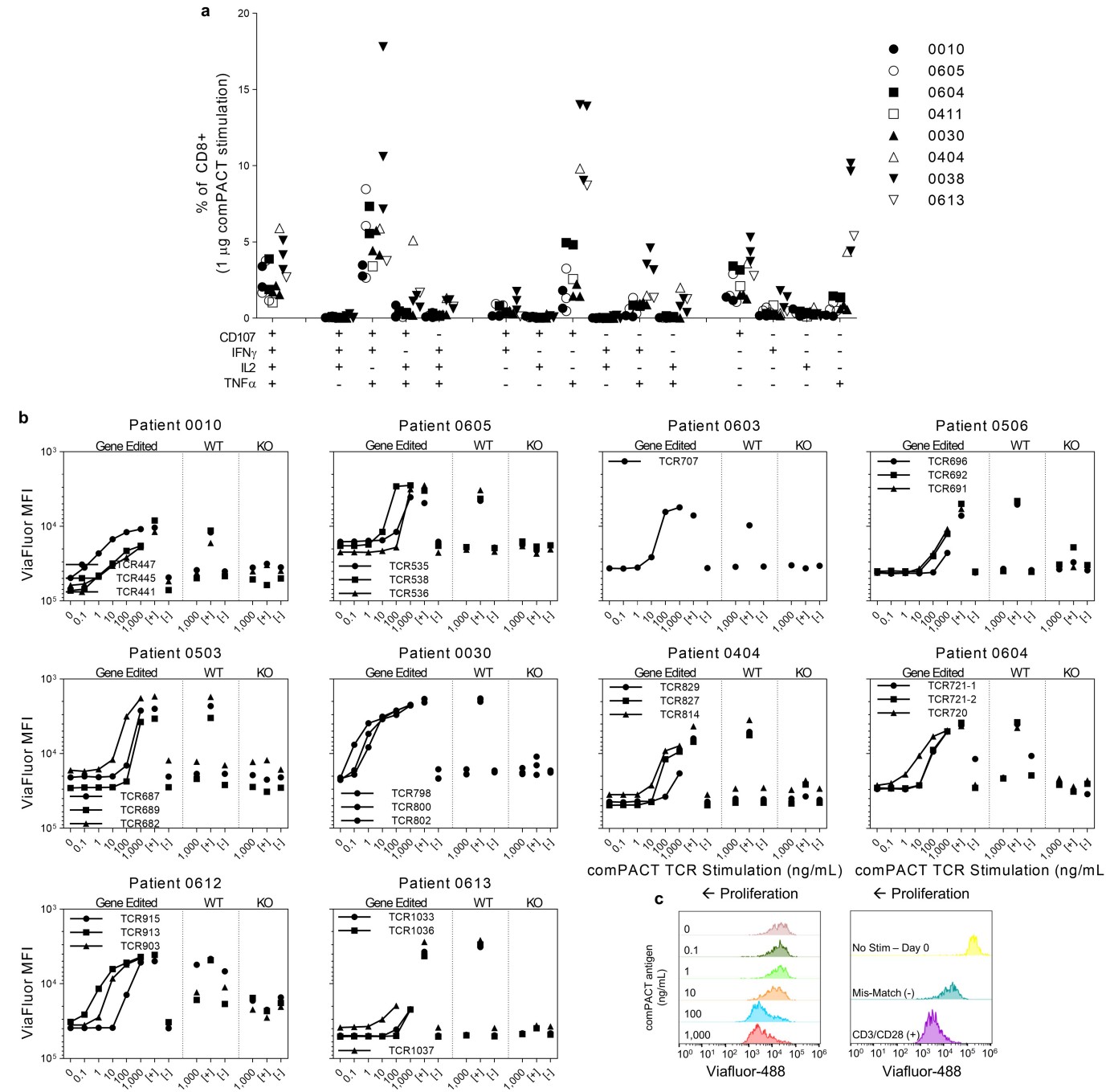

**Extended Data Fig. 2 | Functionality of neoTCR engineered T cells.**
**a**) Intracellular cytokine staining upon activation with cognate peptide-HLA. NeoTCR T cells produce a polyfunctional cytokine profile on antigen encounter. Cells from the clinical final cell product were stimulated overnight with plate-bound peptide-HLA. Percent of CD8 cells positive for the given markers is shown. **b** and **c**) T cells were stained with Viafluor membrane bound dye, stimulated with plate-bound peptide-HLA overnight, and proliferation measured 4 days later. **b**) Concentration-dependent proliferation of individual patient products. CD3/CD28 stimulation positive control indicated by [+] and mis-match compact negative control (used at 1000 ng/mL) indicated by [−]. Y-axis inverted; increased proliferation has lower ViaFluor MFI signal due to dilution of the dye after cell division. **c**) Leftward shift in the Viafluor MFI indicates increasing proliferation (left). Mis-matched peptide-HLA served as a negative control, and CD3/CD28 stimulation as a positive control (right).

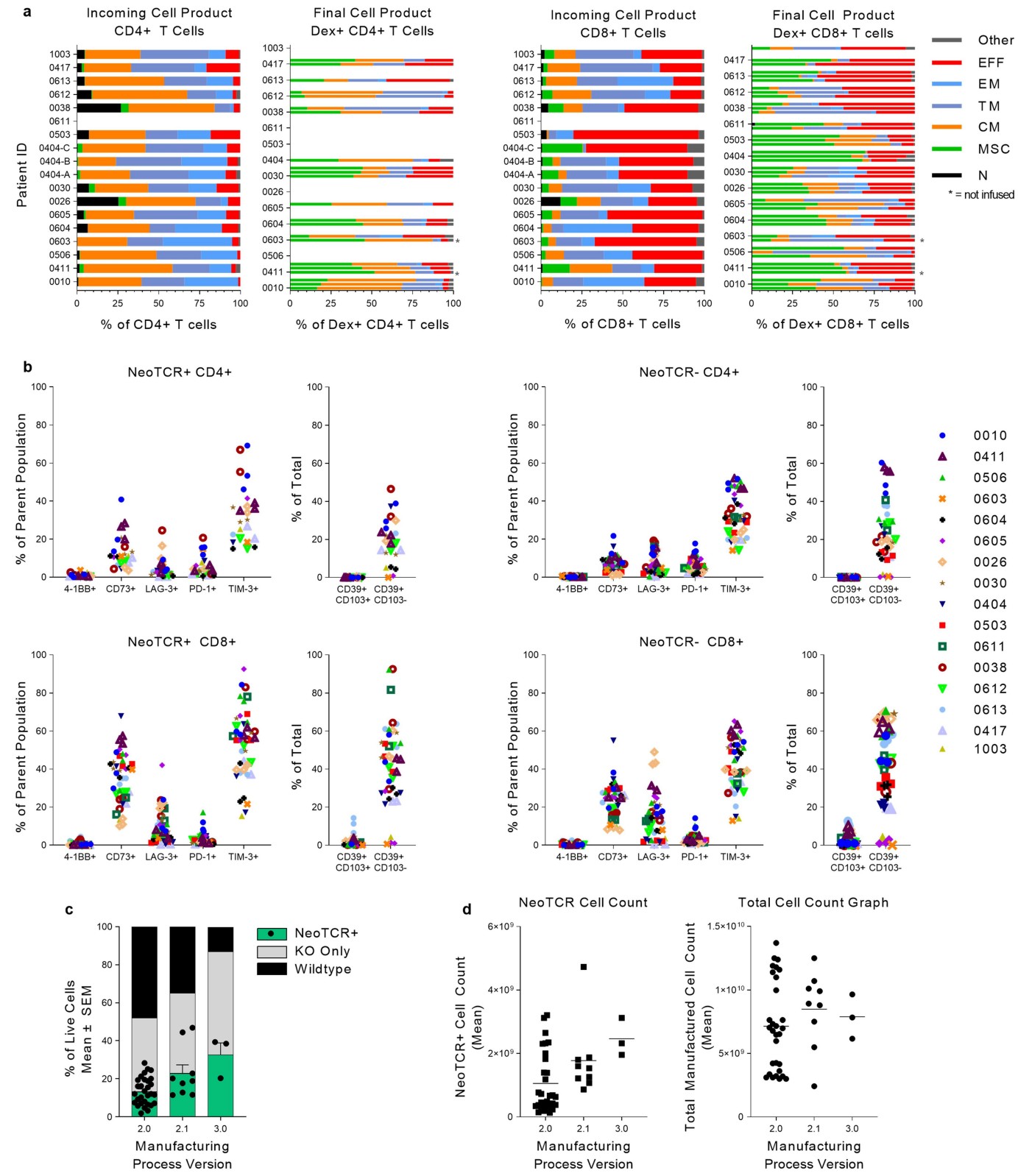

**Extended Data Fig. 3 |** See next page for caption.

**Extended Data Fig. 3 | Characteristics of the manufactured product.**
**a**) Phenotype of CD4+ T cells (left) and CD8+ T cells (right) in incoming leukapheresis and final cell product from dosed patients. Bars represent individual NeoTCR-T cell products for each patient (up-to-3 neoTCRs per patient). For Dex+ CD4+ T cells, only products where the peptide-HLA multimer binds the inserted TCR in the absence of the CD8 co-receptor have data. T cell subset abbreviations are as follows: EFF (effector), EM (effector memory), TM (transitional memory), CM (central memory), MSC (memory stem cell), N (naïve). **b**) T cell activation and phenotypic markers in the manufactured FCP. Percentage of CD4+ (top) or CD8+ (bottom) NeoTCR+ (left) or NeoTCR- (right) cells in the manufactured product that express the indicated surface markers. For NeoTCR+ CD4+ T cells, only products where the dextramer binds the inserted TCR in the absence of the CD8 co-receptor have data. **c**) NeoTCR knock-in efficiency of the endogenous TCR improved with changes in the manufacturing process. NeoTCR+ percentages were significantly different with the different process versions (***p = 0.0006 by ANOVA; v2.1 and v3.0 were significantly better than process v2.0: *p = 0.0218 and **p = 0.0029, respectively, by Tukey's multiple comparisons test, v2.0: n = 30, v2.1: n = 9, v3.0, n = 3). **d**) Cell counts of neoTCR+ cells (left) and total cells (right) in manufacturing process v2.0 (n = 30) compared to process v2.1 (n = 9) and v3.0 (n = 3). Differences not significant (ns) by one-way ANOVA.

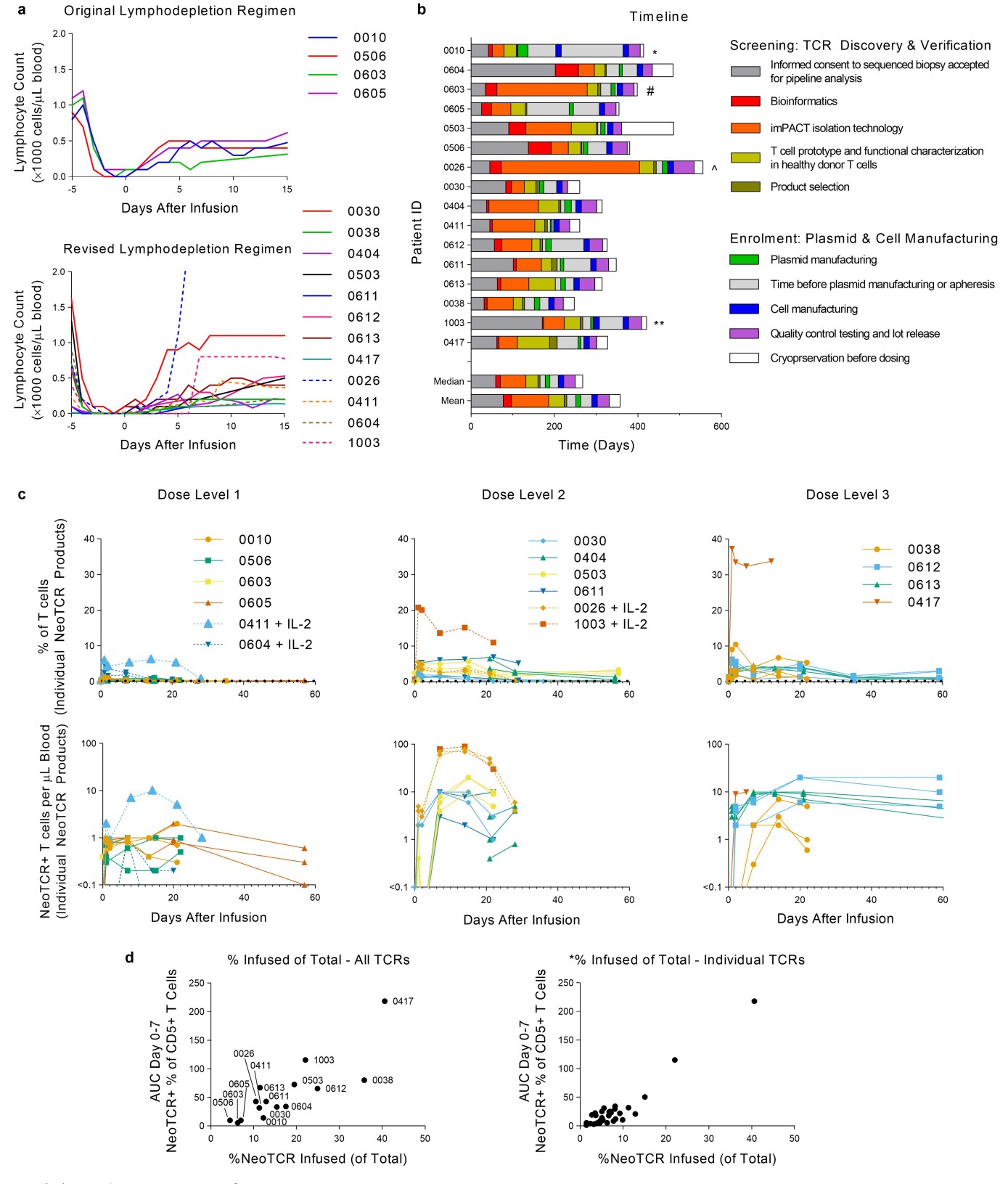

**Extended Data Fig. 4** | See next page for caption.

**Extended Data Fig. 4 | Engineered neoTCR T cell delivery to patients.**
**a**) Absolute lymphocyte counts according to the original conditioning chemotherapy regimen (top), or the revised conditioning chemotherapy regimen (bottom). Patients treated with IL-2 combination therapy are indicated by dotted lines. **b**) Time to generate neoTCR T cell product for 16 dosed patients, ordered by consent date. * 0010: Due to COVID-19 shutdown in 2020 and updates to the manufacturing process, the patient underwent two apheresis and two manufactures of cell therapy products. # 0603: NeoTCR isolation was done three times for repeated attempts to find neoTCRs available for product selection. ^ 0026: Went through five separate PBMC samples before suitable neoTCRs were identified for product selection. ** 1003: Went through two manufactures of the cell therapy product. **c**) NeoTCR percentage (top) and counts (bottom) by dose level, separated by individual neoTCR (up to 3 per patient). Peripheral blood analysis of neoTCR cells in patients treated with dose level 1 (left), dose level 2 (centre), and dose level 3 (right). Total number of neoTCR cells was calculated per μL of blood per patient. Count information was not available for all timepoints. Patients treated with IL-2 are shown with dotted lines. **d**) Gene editing efficiency of final cell product correlates with neoTCR+ cells detected post-infusion. Percent of neoTCR+ cells infused in each patient (left; correlation Pearson r = 0.8463, ****P < 0.0001). Percent of neoTCR+ cells infused per TCR (right; correlation Spearman r = 0.7475, ****P < 0.0001). Area under the curve (AUC) was calculated from day 0 (pre-infusion) up to day 7. Data not shown for patient 0404; no day 0–7 post-infusion samples available.

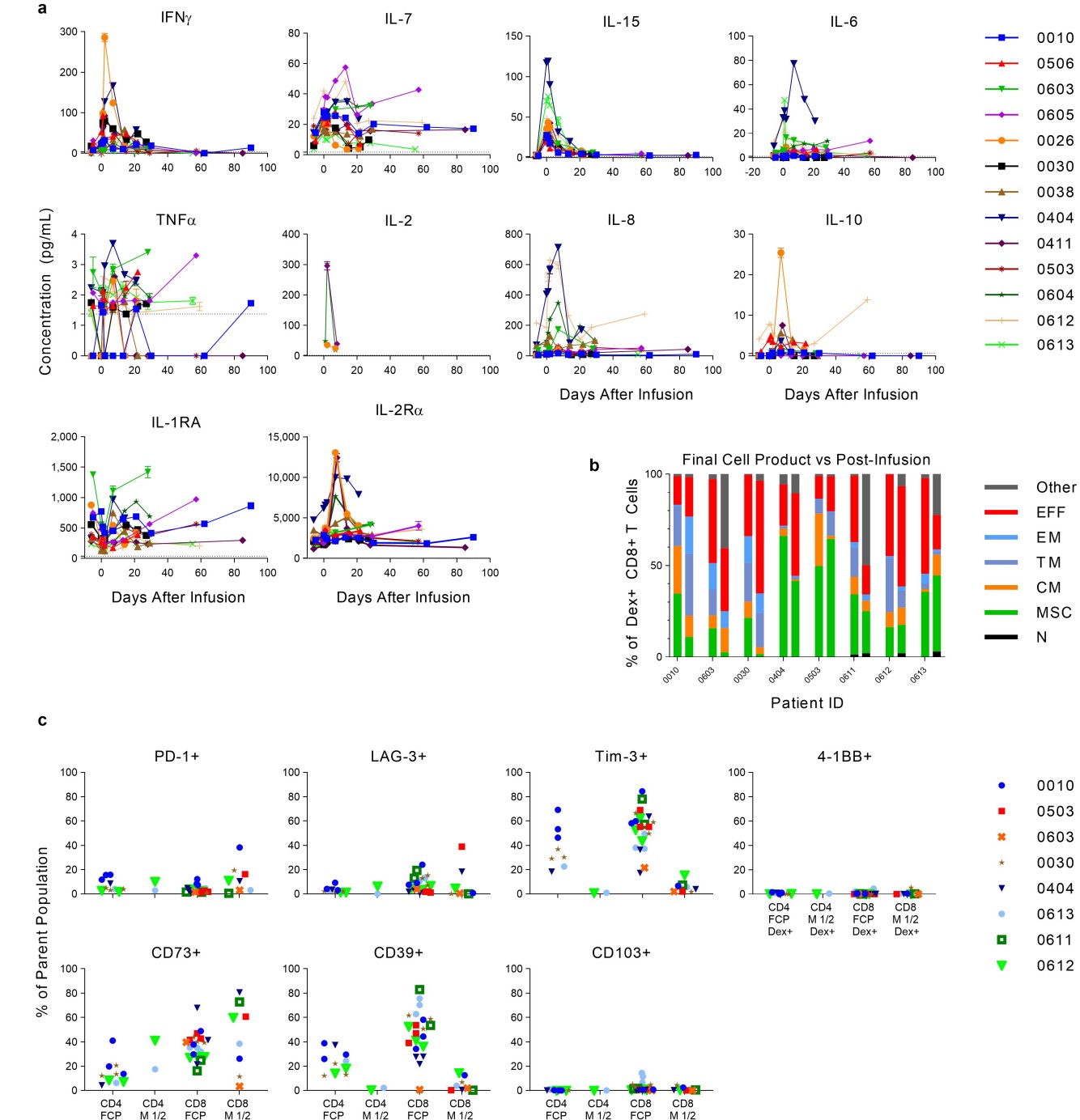

**Extended Data Fig. 5 | Post-infusion analysis of T cells in peripheral blood and serum cytokines. a)** Serum cytokine levels measured using the MSD electrochemiluminescence platform. Thirteen cytokines were measured longitudinally. Horizontal dotted lines represent the lower limit of quantification (LLOQ). IL-12 p70 (0411), IL-13 (0612), and GM-CSF (0038) were below the LLOQ for all but one patient (listed in parenthesis) and are not shown. IL-2 was detected only in patients treated with IL-2 combination therapy (0604, 0411, 0026). Samples measured but below LLOQ are entered as 0. No data for patients 0611, 0417 and 1003. **b)** Analysis of T cell phenotype of the final product and TCR transgenic cells recovered from blood of patients. Phenotype of dextramer+ CD8+ T cells in final cell product (left bars) compared to post-dose samples at 1-2 months after infusion (right bars). Final cell product phenotype shown here is the average of all the patients' products. **c)** T cell activation and phenotypic markers in the manufactured product compared to month 1-2 post-dose for a subset of patients. CD4 (left two columns) and CD8 (right two columns) are shown separately.

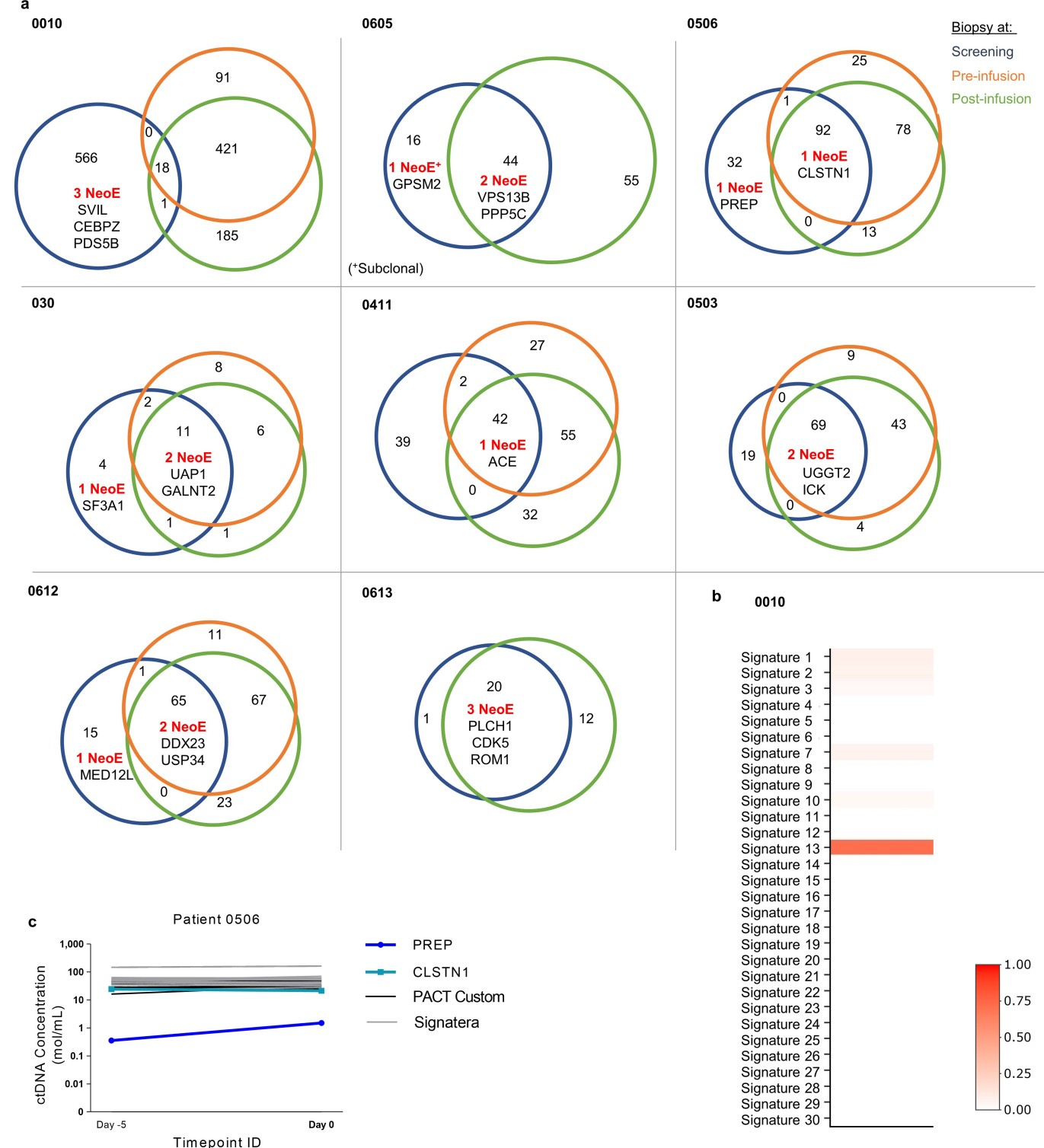

**Extended Data Fig. 6 | Longitudinal retrospective analysis of epitope persistence, somatic signatures and ctDNA data. a**) Venn diagrams of patients with longitudinal screening, pre and post infusion biopsies where available showing protein altering mutation overlap and targeted neoantigen persistence patterns. **b**) Somatic signature analysis of somatic exome mutations and their correlation with known somatic signatures in the COSMIC database.

Signature 13 has previously been associated with APOBEC activity. **c**) Bespoke ctDNA assay for patient 0506 at day −5 and day 0 timepoints showing truncal mutations in gray and targeted neoantigens in aqua and blue respectively. The PREP neoantigen (blue) is detectable by ctDNA and is at lower ctDNA concentrations than predicted truncal mutations.

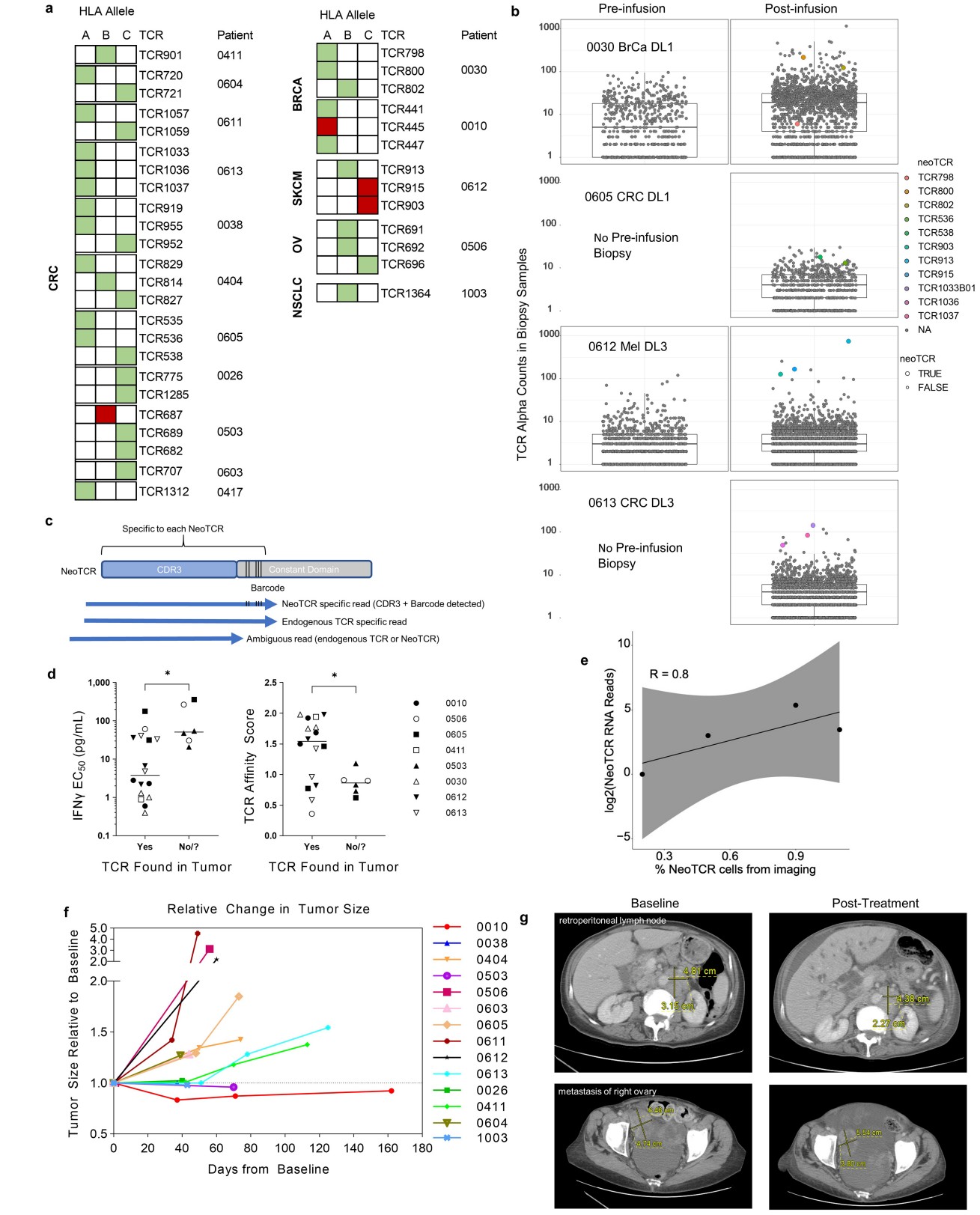

**Extended Data Fig. 7 |** See next page for caption.

**Extended Data Fig. 7 | Tumour biopsy analyses and clinical responses.**
**a**) Retrospective analysis for HLA loss of heterozygosity (LOH, red fill) or no LOH (green fill). Each row is a neoTCR and columns are the HLA allele presenting its targeted epitope (A, B, C). TCGA study codes for the patients' tumor type are shown on the left. **b**) TCR and neoTCR CDR3 quantification in baseline and post-infusion biopsies. Absolute TCRα CDR3 reads from TCR assay were plotted. BrCa: Breast Cancer, CRC: Colorectal Cancer, Mel: melanoma; DL1: Dose-level 1, DL3: Dose-level 3. Boxes indicate the interquartile range (IQR); centre line, median; whiskers, lowest and highest values within 1.5x IQR from the first and third quartiles, respectively. **c**) Schematic of the neoTCR CDR3 and its flanking barcode sequence that can be used to identify endogenous TCR or neoTCR specific reads. **d**) TCRs with a lower IFNγ $EC_{50}$ at lot release (left, *p = 0.0265) or higher TCR affinity score (right, *p = 0.0152) were more frequently found in the post-infusion biopsy. Centre line is the median; p-value by un-paired two-tailed t-test. n = 22; 16 found in the tumour, 6 not identified. For patient 0503, only the specific neoTCR sequence could not be determined. **e**) Correlation of percent of neoTCR cells from imaging versus corresponding sum of neoTCR CDR3 reads detected in post-infusion biopsies. Shaded grey represents the 95% confidence interval. The Pearson correlation coefficient was 0.8. **f**) Spider plot of the change in the sum of each patient's index lesions over time, relative to the baseline scan. No tumour assessment data for patient 0030 (skin lesions) or 0417. **g**) Computed tomography scans for patient 1003 at baseline (day −12, left panel) and on treatment (day 30, right panel).

**Extended Data Table 1 | Mutational load, predicted neoantigen peptide-HLA capture reagents, recognized mutations and TCR clonotypes for the 16 patients infused in the clinical trial. TMB, tumour mutational burden and was calculated as follows: TMB=#NSM/35 MB, where 35 MB is the length of the sequencing footprint**

| Dose level | Patient ID | # NSM (TMB) | # expressed mutations | HLA alleles covered by HLA library | # HLA-neoantigen capture reagents proposed* | # HLA-neoantigen capture reagents produced | # recognized neoantigens[†] | # unique TCRs isolated[‡] | # unique TCRs confirmed[§] |
|---|---|---|---|---|---|---|---|---|---|
| **DL 1** | 0010 | 468 (13.4) | 236 | A*02:01, A*24:02, B*35:02, C*12:03 | 352 | 262 | 6 | 15 (11) | 6 |
| | 0605 | 60 (1.7) | 23 | A*02:01, A*24:02, C*01:01, C*03:03 | 86 | 49 | 4 | 6 | 3 |
| | 0603 | 202 (5.8) | 88 | A*26:01, B*42:01, B*44:02, C*05:01, C*17:01 | 288 | 66 | 3 | 9 (3) | 2 |
| | 0506 | 125 (3.6) | 56 | A*24:02, B*35:01, B*46:01, C*01:02, C*03:03 | 352 | 105 | 3 | 5 | 5 |
| **DL 2** | 0503 | 88 (2.5) | 30 | A*24:02, B*39:01, B*52:01, C*07:02, C*12:02 | 352 | 130 | 5 | 9 (8) | 4 |
| | 0030 | 29, 31 (0.9) | 20 | A*02:01, A*11:01, B*35:01, C*04:01 | 352 | 117 | 7 | 10 (8) | 7 |
| | 0404 | 120 (3.4) | 35 | A*01:01, A*31:01, B*08:01, B*40:01, C*03:04, C*07:01 | 352 | 94 | 3 | 7 (6) | 6 |
| | 0611 | 74 (2.1) | 25 | A*01:01, A*24:02, B*57:01, C*04:01, C*06:02 | 352 | 67 | 4 | 8 | 4 |
| **DL 3** | 0038 | 95 (2.7) | 34 | A*02:01, A*24:02, B*07:02, B*51:01, C*15:02, C*07:02 | 352 | 125 | 10 | 30 (16) | 9 |
| | 0612 | 244 (7.0) | 81 | A*01:01, B*08:01, B*07:02, C*07:01, C*07:02 | 352 | 126 | 3 | 16 (14) | 3 |
| | 0613 | 43 (1.2) | 21 | A*02:01, C*07:02 | 352 | 83 | 5 | 8 | 6 |
| | 0417 | 107 (3.1) | 62 | A*02:01, A*25:01, B*15:01, B*18:01, C*03:03, C*12:03 | 352 | 147 | 11 | 17 (10) | 6 |
| **NeoTCR-P1 + IL-2** | 0604 | 102 (2.9) | 30 | A*01:01, A*11:01, B*08:01, B*35:01, C*04:01, C*07:01 | 352 | 98 | 3 | 3 | 2 |
| | 0411 | 83 (2.4) | 32 | A*01:01, A*02:01, B*07:02, B*57:01, C*06:02, C*07:02 | 352 | 87 | 5 | 5 | 3 |
| | 0026 | 89 (2.5) | 48 | A*01:01, B*08:01, C*07:01 | 352 | 35,104 | 6 | 22 (11) | 4 |
| | 1003 | 172 (4.9) | 94 | A*02:01, A*26:01, B*15:01, B*35:01, C*03:04, C*04:01 | 352 | 146 | 5 | 5 | 3 |
| **Median** | | 102 (2.9) | 35 | 5 | 352 | 104 | 5 | 8 | 4 |
| **Total** | | | | 34 (unique) | 5302 | 1841 | 83 | 175 (127) | 73 |

*A maximum of 352 predicted neoantigen capture reagents were provided for protein synthesis.

[†]Number of unique non-synonymous somatic mutations recognized by one or more TCRs isolated from patient PBMCs.

[‡]Number of unique TCRs isolated from the patient PBMCs, number in parenthesis indicates the numbers that were passed on for confirmation, if it was less than the total number of unique TCRs isolated.

[§]Number of unique TCRs that were transfected into healthy donor cells and showed specific binding to the matched peptide-HLA and IFNγ secretion with peptide-HLA stimulation.

drome; SD: Stable Disease; PD: Progressive Disease; Y/N: Yes/No.

**Extended Data Table 2 | Targeted neoantigen, manufacturing process, neoTCR transfection efficiency and function for each clinical infused product to the 16 patients in this cohort**

| Dose level | Patient ID | Cell manufacturing process version | Gene | mRNA transcript per million (TPM) at screen | HLA Allele | Selected TCR ID | IFNγ EC$_{50}$ (at lot release, pg/mL)[*] | NeoTCR+ T cells (as % of live)[*] |
|---|---|---|---|---|---|---|---|---|
| DL1 | 0010 | 2.0 | SVIL | 2.8 | A*02:01 | TCR441 | 5 | 11.6 |
| | | | CEBPZ | 2.3 | A*24:02 | TCR445 | 1 | 12.1 |
| | | | PDS5B | 0.6 | A*02:01 | TCR447 | 101 | 16.2 |
| | 0605 | 2.0 | VPS13B | 5 | A*02:01 | TCR535 | 362 | 9.3 |
| | | | PPP5C | 52 | A*24:02 | TCR536 | 177.9 | 7.4 |
| | | | GPSM2[†] | 271 | C*03:03 | TCR538 | 31.6 | 6.0 |
| | 0603 | 2.0 | TENM4 | 1 | C*05:01 | TCR707 | 39.8 | 6.5 |
| | 0506 | 2.0 | PREP | 24 | B*35:01 | TCR692 | 61.5 | 1.9 |
| | | | CLSTN1 | 45 | B*35:01 | TCR691 | 30.9 | 10.2 |
| | | | CLSTN1 | 45 | C*03:03 | TCR696 | 266.4 | 17 |
| DL2 | 0503 | 2.0 | UGGT2 | 61 | C*12:02 | TCR682 | 20.8 | 19.6 |
| | | | ICK | 36 | C*12:02 | TCR689 | 48.1 | 19.3 |
| | | | ICK | 36 | B*39:01 | TCR687 | 54.6 | 21.3 |
| | 0030 | 2.0 | UAP1 | 25 | A*02:01 | TCR798 | 1 | 12 |
| | | | GALNT2 | 29 | B*35:01 | TCR802 | 1.3 | 16.1 |
| | | | SF3A1 | 5 | A*02:01 | TCR800 | 0.4 | 15.4 |
| | 0404 | 2.0 | STRADA | 9 | A*01:01 | TCR829 | 38.9 | 25.2 |
| | | | MTHFR | 51 | B*40:01 | TCR814 | 6.2 | 20.1 |
| | | | STRADA | 9 | C*07:01 | TCR827 | 13 | 28.3 |
| | 0611 | 2.0 | CDAC1 | 10 | A*24:02 | TCR1057 | 82.6 | 4.7 |
| | | | FCGRT[†] | 1 | C*06:02 | TCR1059 | 35.7 | 18.9 |
| DL3 | 0038 | 2.1 | NAALADL2 | 47 | A*02:01 | TCR919 | 1.6 | 46.8 |
| | | | BRPF1 | 8 | A*24:02 | TCR955 | 5.3 | 44.4 |
| | | | PAPSS1 | 39 | C*07:02 | TCR952 | 1.2 | 20.2 |
| | 0612 | 2.1 | DDX23 | 38 | B*08:01 | TCR913 | 2.2 | 17.6 |
| | | | USP34 | 67 | C*07:02 | TCR903 | 6.7 | 22.9 |
| | | | MED12L | 14 | C*07:02 | TCR915 | 36.3 | 18.9 |
| | 0613 | 2.1 | PLCH1 | 25 | A*02:01 | TCR1036 | 41.2 | 11.4 |
| | | | CDK5 | 18 | A*02:01 | TCR1033 | 33.2 | 12.8 |
| | | | ROM1 | 5 | A*02:01 | TCR1037 | 4.8 | 11.6 |
| | 0417 | 3.0 | RASA3 | 125 | A*02:01 | TCR1312 | 19.2, 25.8 | 40.0, 38.0 |
| NeoTCR-P1 + IL-2 | 0604 | 2.0 | WDFY3 | 14 | A*11:01 | TCR720 | 5.2 | 11.2 |
| | | | MAN2A2 | 132 | C*04:01 | TCR721 | 90.9 | 23.2, 24.6[‡] |
| | 0411 | 2.0 | ACE | 44 | B*57:01 | TCR901 | 0.9 | 13.5, 8.0, 3.2[‡] |
| | 0026 | 2.0 | ETNK1 | 159 | C*07:01 | TCR1285 | 23.7, 11.2 | 5.7, 7.1 |
| | | | PRCC | 85 | C*07:01 | TCR775 | 13.4 | 7.1 |
| | 1003 | 3.0 | KBTBD3 | 18 | B*35:01 | TCR1364 | 98.7[§] | 20.3 |

[*] Multiple values listed when more than one cell product lot was manufactured with the same TCR.
[†] Predicted subclonal mutation.
[‡] Lot not infused; intended dose level reached without infusion.
[§] IFNγ measured using the ELLA Simple Plex; all other reported values measured by IFNγ ELISA

**Extended Data Table 3 | Patient and disease characteristics, adverse events and response assessment**

| Dose level | Patient ID | Age | Cancer | # Prior regimens | # TCRs | Conditioning regimen | Total NeoTCR+ cell dose | Any AEs ≥ grade 3 and SAEs | TCR-related AEs | Response |
|---|---|---|---|---|---|---|---|---|---|---|
| DL 1 | 0010 | 38 | HR+ Breast | 7 | 3 | Cy 300 mg/m² x3d Flu 30 mg/m² x3d | $4 \times 10^8$ | G3 neutropenia | | SD (target lesions ↓ 17%) for 4m |
| | 0605 | 53 | MSS-CRC | 4 | 3 | Cy 300 mg/m² x3d Flu 30 mg/m² x3d | $4 \times 10^8$ | SAE D52 small bowel obstruction | | PD |
| | 0603 | 65 | MSS-CRC | 4 | 1 | Cy 300 mg/m² x3d Flu 30 mg/m² x3d | $2 \times 10^8$ | COVID-19 + pneumonia | | PD |
| | 0506 | 70 | Ovarian | 6 | 3 | Cy 300 mg/m² x3d Flu 30 mg/m² x3d | $4 \times 10^8$ | G3 neutropenia | | PD |
| DL 2 | 0503 | 48 | MSS-CRC | 9 | 3 | Cy 600 mg/m² x3d Flu 30 mg/m² x4d | $1.3 \times 10^9$ | G4 neutropenia SAE: UTI D74 | | PD |
| | 0030 | 45 | HR+ Breast | 5 | 3 | Cy 600 mg/m² x3d Flu 30 mg/m² x4d | $1.3 \times 10^9$ | G4 neutropenia | | SD at D28 and D56 |
| | 0404 | 47 | MSS-CRC | 7 | 3 | Cy 600 mg/m² x3d Flu 30 mg/m² x4d | $1.3 \times 10^9$ | G4 neutropenia SAE G3 Peri-hepatic hematoma | | PD |
| | 0611 | 44 | MSS-CRC | 5 | 2 | Cy 600 mg/m² x3d Flu 30 mg/m² x4d | $9 \times 10^8$ | G2 Headaches week 2 | | PD |
| DL 3 | 0038 | 39 | MSS-CRC | 2 | 3 | Cy 600 mg/m² x3d Flu 30 mg/m² x4d | $4 \times 10^9$ | G4 neutropenia | | PD |
| | 0612 | 47 | Melanoma | 3 | 3 | Cy 600 mg/m² x3d Flu 30 mg/m² x4d | $4 \times 10^9$ | G4 neutropenia | | PD |
| | 0613 | 36 | MSS-CRC | 3 | 3 | Cy 600 mg/m² x3d Flu 30 mg/m² x4d | $4 \times 10^9$ | SAE: G4 febrile neutropenia | G1 CRS | SD at D28 and D56 |
| | 0417 | 38 | MSS-CRC | 4 | 1 | Cy 600 mg/m² x3d Flu 30 mg/m² x4d | $5.4 \times 10^9$ | SAE: G4 Hyponatremia SAE: G5 Malignant neoplasm progression | | No post-baseline assessment |
| NeoTCR-P1 + IL-2 | 0604 | 40 | MSS-CRC | 5 | 2 | Cy 600 mg/m² x3d Flu 30 mg/m² x4d | $4 \times 10^8$ + IL-2 | G3 neutropenia and febrile neutropenia; SAE: G3 pancreatitis D40 | | PD |
| | 0411 | 58 | MSS-CRC | 5 | 1 | Cy 600 mg/m² x3d Flu 30 mg/m² x4d | $7.5 \times 10^8$ + IL-2 | G4 neutropenia | | SD at D28 and D56 |
| | 0026 | 58 | MSS-CRC | 5 | 2 | Cy 600 mg/m² x3d Flu 30 mg/m² x4d | $1.3 \times 10^9$ + IL-2 | G4 neutropenia | | PD |
| | 1003 | 68 | NSCLC | 3 | 1 | Cy 600 mg/m² x3d Flu 30 mg/m² x4d | $1.96 \times 10^9$ + IL-2 | G3 encephalopathy | G3 encephalopathy | SD at D28 |

MSS-CRC: Microsatellite Stable Colorectal Cancer; HR: Hormone Receptor; G: Grade; SAE: Serious Adverse Event; CRS: Cytokine Release Syndrome; SD: Stable Disease; PD: Progressive Disease; Y/N: Yes/No.

# Reporting Summary

## Statistics

For all statistical analyses, confirm that the following items are present in the figure legend, table legend, main text, or Methods section.

| n/a | Confirmed | |
|---|---|---|
| ☐ | ☒ | The exact sample size (*n*) for each experimental group/condition, given as a discrete number and unit of measurement |
| ☐ | ☒ | A statement on whether measurements were taken from distinct samples or whether the same sample was measured repeatedly |
| ☐ | ☒ | The statistical test(s) used AND whether they are one- or two-sided *Only common tests should be described solely by name; describe more complex techniques in the Methods section.* |
| ☒ | ☐ | A description of all covariates tested |
| ☐ | ☒ | A description of any assumptions or corrections, such as tests of normality and adjustment for multiple comparisons |
| ☐ | ☒ | A full description of the statistical parameters including central tendency (e.g. means) or other basic estimates (e.g. regression coefficient) AND variation (e.g. standard deviation) or associated estimates of uncertainty (e.g. confidence intervals) |
| ☐ | ☒ | For null hypothesis testing, the test statistic (e.g. *F*, *t*, *r*) with confidence intervals, effect sizes, degrees of freedom and *P* value noted *Give P values as exact values whenever suitable.* |
| ☒ | ☐ | For Bayesian analysis, information on the choice of priors and Markov chain Monte Carlo settings |
| ☒ | ☐ | For hierarchical and complex designs, identification of the appropriate level for tests and full reporting of outcomes |
| ☐ | ☒ | Estimates of effect sizes (e.g. Cohen's *d*, Pearson's *r*), indicating how they were calculated |

*Our web collection on statistics for biologists contains articles on many of the points above.*

## Software and code

Policy information about availability of computer code

| Data collection | Flow cytometry data was collected using BD FACSDiva (V8.0.3) and analysed with FlowJo (V10.7.1 or V10.8.1), or FCS Express (V6.6.21.0). Serum cytokine analysis was performed using a MESO QuickPlex SQ 120 instrument and Discovery Workbench 4.0 software. |
|---|---|
| Data analysis | The following software was used for analysis (Version and or location listed in parenthesis): OptiType (1.3.4), netMHCpan (3.0, 4.0, or 4.1 as indicated in the methods), RSEM (1.3.3), STAR (2.7.6a), MiXCR (2.1.3), VarDictJava (1.8.2), VarScan (2.4.4), Sentieon (BWA, 201911.01), Sequenza (3.0), Strelka2 (2.9.10), Mutect (3.1-0-g72492bb), MuTect2 (4.1.8.1), pyClone (0.13.1), Ensembl (release 101, http://ensembl.org/), GraphPad Prism (9.4.1). |

For manuscripts utilizing custom algorithms or software that are central to the research but not yet described in published literature, software must be made available to editors and reviewers. We strongly encourage code deposition in a community repository (e.g. GitHub). See the Nature Portfolio guidelines for submitting code & software for further information.

## Data

Policy information about availability of data

All manuscripts must include a data availability statement. This statement should provide the following information, where applicable:
- Accession codes, unique identifiers, or web links for publicly available datasets
- A description of any restrictions on data availability
- For clinical datasets or third party data, please ensure that the statement adheres to our policy

The following publicly available data sets were utilised: ExAc (3.1, https://gnomad.broadinstitute.org/downloads#exac-variants), dbSNP (v146, ftp://ftp.broadinstitute.org/bundle), GATK Resource Bundle (hg19/Grch37, ftp://ftp.broadinstitute.org/bundle), Human Proteome (Homo_sapiens.GRCh37.75.pep.all.fa, http://ensembl.org/), IMGT (TCR/HLA, 3.1.17, http://www.imgt.org/), RefSeq (1052019, ftp://hgdownload.cse.ucsc.edu/goldenPath), TCGA (Version 1.0, https://portal.gdc.cancer.gov/), Broad Institute (hg19, ftp://ftp.broadinstitute.org/bundle). The TCR sequences from the present study are available in the article supplemental files, and the genomics data is available on reasonable request from the European Genome-Phenome Archive (EGA) repository.

## Human research participants

Policy information about studies involving human research participants and Sex and Gender in Research.

| Reporting on sex and gender | Patients were screened and enrolled on this study irrespective of their sex/gender. Any data regarding a patient's sex and gender was collected and provided to the sponsor by the treating physician and PI for the study at each site. No sex- or gender-based analysis has been conducted in this small dataset. |
|---|---|
| Population characteristics | Provided in extended data table 3. |
| Recruitment | Patients were recruited across 9 clinical investigational sites. Given the phase 1 nature and complexity of the study, the sites were limited to the United States of America. There were no biases introduced and patients were screened on a first-come first-serve basis based on meeting the protocol inclusion-exclusion criteria. The principal investigators identified patients based on the inclusion exclusion criteria and contacted the PACT medical monitor to confirm if an informed consent form could be signed. No protocol waivers were allowed on this study. |
| Ethics oversight | The trial was conducted in accordance with the principles of the Declaration of Helsinki. The trial protocol and statistical analysis plan were designed in a collaboration between the sponsor (PACT Pharma Inc.) and the authors. The protocol was approved by the institutional review board from each clinical site enrolling patients: City of Hope, Duarte California; University of California Los Angeles, Los Angeles California; University of California, Irvine Medical Center, Orange, California; University of California, Davis, Sacramento California; University of California, San Francisco, San Francisco California; Northwestern University Medical Center, Chicago Illinois; Memorial Sloan Kettering Cancer Center, New York, New York; Tennessee Oncology, Nashville, Tennessee; and Fred Hutchinson Cancer Research Center, Seattle, Washington. |

Note that full information on the approval of the study protocol must also be provided in the manuscript.

# Field-specific reporting

Please select the one below that is the best fit for your research. If you are not sure, read the appropriate sections before making your selection.

☒ Life sciences          ☐ Behavioural & social sciences          ☐ Ecological, evolutionary & environmental sciences

For a reference copy of the document with all sections, see nature.com/documents/nr-reporting-summary-flat.pdf

# Life sciences study design

All studies must disclose on these points even when the disclosure is negative.

| Sample size | Up to approximately 76 evaluable participants will be enrolled into the Initial Phase. The planned enrollment for the Expansion Phase study is potentially up to 112 participants, depending on the number and size of the cohorts. The total anticipated enrollment in this study is approximately 9–188 participants.<br>Three to 12 participants will be enrolled into each dose level cohort in the Phase 1a portion of the study. If the study proceeds to the dose-expansion basket cohorts in the Phase 1a, up to 40 additional participants may be enrolled (up to 20 each in the TCR alone and TCR + IL-2 baskets).<br>The dose-escalation stage sample size was based on the probability of not observing any DLTs in 3 participants, and the probability of observing fewer than 2 DLTs in 6 participants for underlying DLT rates during the dose-escalation stage. |
|---|---|
| Data exclusions | Patients went through a screening process for TCR selection and cell therapy manufacture. Patients were excluded from dosing if they failed eligibility criteria or if a product could not be manufactured. Only products manufactured using version 2.0 or 2.1 were included in the analysis. No primary or secondary endpoint data were excluded from the analysis. All available manufactured products and PBMCs were analysed. Two post-infusion biopsies were excluded from analysis due to insufficient tumour content. |

| | |
|---|---|
| Replication | The NeoTCR-P1 is an autologous TCR therapy manufactured with the patients' own PBMCs and their own unique TCRs. The cell therapies were manufactured with up to three independent lots consisting of three unique TCRs or, in some cases, a single TCR, giving technical replicates of the manufacturing for an individual patient. The study included treatment of human participants with a personalized NeoTCR-P1 cell therapy product. Due to the disease characteristics and personalized nature of the cell therapy product, replication of the findings may vary depending on the disease state and the neoTCR selected for infusion. Flow cytometry experiments to analyse the final cell product or post-infusion PBMC samples were performed in duplicate, if there were enough cells available, and all attempts at replication were successful. |
| Randomization | This was a Phase 1a, open label, 3+3 dose escalation trial design trial design to evaluate NeoTCR-P1 infused as a single agent without or with IL-2, or in combination with nivolumab. Patients were not randomized and were enrolled at the maximum open dose level during the trial, if enough cells were manufactured to meet a given dose level. Once a dose level was cleared, patients and their treating physician had the option to administer the NeoTCR-P1 cell therapy in combination with IL-2. No patients were treated in combination with nivolumab. |
| Blinding | This was a single arm, open label trial design, thus blinding is not relevant to the study. |

# Reporting for specific materials, systems and methods

We require information from authors about some types of materials, experimental systems and methods used in many studies. Here, indicate whether each material, system or method listed is relevant to your study. If you are not sure if a list item applies to your research, read the appropriate section before selecting a response.

## Materials & experimental systems

| n/a | Involved in the study |
|---|---|
| ☐ | ☒ Antibodies |
| ☐ | ☒ Eukaryotic cell lines |
| ☒ | ☐ Palaeontology and archaeology |
| ☒ | ☐ Animals and other organisms |
| ☐ | ☒ Clinical data |
| ☒ | ☐ Dual use research of concern |

## Methods

| n/a | Involved in the study |
|---|---|
| ☒ | ☐ ChIP-seq |
| ☐ | ☒ Flow cytometry |
| ☒ | ☐ MRI-based neuroimaging |

## Antibodies

| | |
|---|---|
| Antibodies used | The antibodies used for flow cytometry are detailed in Supplementary Information Table 4. Tumour tissue sections were stained with anti-CD3 (clone EP4426, Abcam; anti-rabbit AF647, ThermoFisher), Vector2A RNAScope Probe to identify neoTCR edited cells (Advanced Cell Diagnostics; Opal 570, Akoya Biosciences, Marlborough, MA), and DAPI (ACD). |
| Validation | Antibodies used for flow cytometry were validated using human PBMCs, isolated T cells, or activated T cells (activated with TransACT for 48-72 h). Antibodies were titrated and fluorescence minus one (FMO) controls were created to set gates for positive events. For tumour tissue staining, anti-CD3 was protein A purified and validated for IHC on Jurkat (Human T cell leukemia T lymphocytes) cells by the manufacturer (Abcam). The Vector2A RNAScope Probe was validated on FFPE neoTCR edited cell pellets, with un-edited cells used as a negative control. |

## Eukaryotic cell lines

Policy information about cell lines and Sex and Gender in Research

| | |
|---|---|
| Cell line source(s) | The SW620 colorectal cancer cell line was purchased from ATCC, and a master cell bank was generated. Cells were transduced to express nucLight red, and further edited to insert an R20Q point mutation in COX6C. Cells were again expanded to generate additional working cell banks. |
| Authentication | Genotyping confirmed editing but cell lines were not further authenticated. |
| Mycoplasma contamination | All cell banks tested negative for mycoplasma. |
| Commonly misidentified lines (See ICLAC register) | Name any commonly misidentified cell lines used in the study and provide a rationale for their use. |

## Clinical data

Policy information about clinical studies

All manuscripts should comply with the ICMJE guidelines for publication of clinical research and a completed CONSORT checklist must be included with all submissions.

| | |
|---|---|
| Clinical trial registration | NCT03970382 |
| Study protocol | The study protocol, "A Phase 1a/1b, Open-label First-in-human Study of the Safety, Tolerability and Feasibility of Gene-edited Autologous NeoTCR-T Cells (NeoTCR-P1) Administered as a Single Agent or in Combination With Anti-PD-1 to Patients With Locally Advanced or Metastatic Solid Tumors" is provided in the Supplemental Information Files. |

| Data collection | From December 2019 to February 2022, the study was active at 9 investigational sites: City of Hope, Duarte California; University of California Los Angeles, Los Angeles California; University of California, Irvine Medical Center, Orange, California; University of California, Davis, Sacramento California; University of California, San Francisco, San Francisco California; Northwestern University Medical Center, Chicago Illinois; Memorial Sloan Kettering Cancer Center, New York, New York; Tennessee Oncology, Nashville, Tennessee; and Fred Hutchinson Cancer Research Center, Seattle, Washington. All samples were collected at the patients' investigational site. Data was analysed at PACT Pharma. |
|---|---|
| Outcomes | Primary Outcomes:<br>1. Incidence of adverse events as defined as Dose limiting toxicity (DLT): DLT was defined as protocol-defined adverse events that occur within 28 days following infusion of Neo-TCR-P1 administered as a single agent or in combination with nivolumab.<br>2. Number of participants with adverse events as a measure of safety and tolerability of NeoTCR-P1 or NeoTCR-P1 in combination with nivolumab: Toxicity was classified and graded according to the National Cancer Institute's Common Terminology Criteria for Adverse Events (CTCAE, version 5.0). Cytokine release syndrome (CRS) and neurotoxicity associated with NeoTCR-P1 will be graded according to ASBMT consensus grading.<br>3. Maximum Tolerated Dose (MTD) of NeoTCR-P1: The MTD was defined as the highest dose with an observed incidence of DLT in no more than one out of six patients treated at a particular dose level.<br>4. Feasibility of manufacturing NeoTCR-P1: Percent of screened patients that enrolled on study and receive NeoTCR-P1<br>Secondary Outcomes:<br>1. Maximum concentration of NeoTCR-P1 (Cmax) in the peripheral blood<br>2. Area-under-the-concentration-vs-time-curve (AUC) in the peripheral blood<br>3. Persistence of NeoTCR-P1 in samples of peripheral blood<br>4. Objective Response Rate (ORR) in participants with solid tumors following infusion of NeoTCR-P1 as a single agent or in combination with nivolumab: ORR was defined as Complete Response (CR) or Partial Response (PR) per RECIST v1.1, as determined by the investigator<br>5. Duration of Response mediated by NeoTCR-P1 administered as a single agent or in combination with nivolumab to participants with solid tumors: Duration of response, defined as time from the first occurrence of a documented objective response to the time of relapse or death from any cause<br>6. Progression free survival (PFS) in participants with solid tumors following infusion of NeoTCR-P1 as a single agent or in combination with nivolumab: PFS will be defined from date of administration of NeoTCR-P1 cell infusion to the date of disease progression per the RECIST v1.1 or death as a result of any cause. Subjects who do not meet criteria for progression by the analysis data cut-off date will be censored at their last evaluable disease assessment date<br>7. Overall survival (OS) in participants with solid tumors following infusion of NeoTCR-P1 as a single agent or in combination with nivolumab: OS will be measured from the date of administration of NeoTCR-P1 to the date of death. Subjects who have not died by the analysis data cut-off date will be censored at their last date of contact. |

# Flow Cytometry

## Plots

Confirm that:

☒ The axis labels state the marker and fluorochrome used (e.g. CD4-FITC).

☒ The axis scales are clearly visible. Include numbers along axes only for bottom left plot of group (a 'group' is an analysis of identical markers).

☒ All plots are contour plots with outliers or pseudocolor plots.

☒ A numerical value for number of cells or percentage (with statistics) is provided.

## Methodology

| Sample preparation | Peripheral Blood Mononuclear Cells (PBMCs) were collected in ACD or CPT tubes and shipped to Precision for Medicine for PBMC isolation and cryopreservation prior to analysis. Apheresis products were obtained from the patient at the study site and shipped overnight to the study sponsor. After cell manufacture, aliquots of T cells were cryopreserved prior to analysis. |
|---|---|
| Instrument | Flow cytometry data was collected on an Attune NxT, or cell sorted using a FACS Aria III. |
| Software | Flow cytometry data was collected using BD FACSDiva (V8.0.8) and analysed with FlowJo (V10.7.1 or V10.8.1), or FCS Express (V6.6.21.0). |
| Cell population abundance | NeoTCR T cells were single-cell sorted from patient PBMCs using two color staining for neoantigen-HLA multimer CD8+ cells, and were detected at a frequency of >1 in 300,000 CD8 T cells. In the incoming cell product, enriched T cells were greater than 90% pure (determined by CD4 and CD8 staining). NeoTCR+ T cell abundance varied with the starting sample but ranged from 1.9-46.8% of live cells in the final cell product and 0.04-37.3% of live cells in post-infusion PBMCs. |
| Gating strategy | Gating strategies are shown in Supplementary Information Section. Fluorescence minus one (FMO) controls were created to set gates for positive events. |

☒ Tick this box to confirm that a figure exemplifying the gating strategy is provided in the Supplementary Information.

