## [Peer Review File · Nature]

Manuscript Title: Non-viral precision T cell receptor replacement for personalized cell therapy

Reviewer Comments & Author Rebuttals

Reviewer Reports on the Initial Version:

Referees' comments:

Referee #1 (Remarks to the Author):

In a phase 1 clinical trial, 14 patients with advanced solid tumors (mostly MSS colorectal Ca) were treated with gene-engineered T cells expressing personalized neoTCRs targeting private cancer mutations present in the tumors from the individual patients.

Personal neoantigens were identified using WES and RNAseq of tumor and normal cells for mutation calling and in silico HLA binding prediction and RNA expression. A median of 88 (range 31-488) mutations and a median of 32 expressed mutations were identified for each patient. Up to 350 neoantigen peptide-HLA candidates across the HLA class I alleles of that patient covered by an HLA library were selected per patient.

By fusing neoantigen peptides to beta-2-microglobulin domains and the HLA, peptide-HLA libraries covering a median of 5 HLAs per patient were then generated (median of 94 peptides per patient). Using DNA-barcoding, fluorescent labelling and multimerization, neoantigen-specific T cells were isolated from CD8+ T cells in the peripheral blood. Across the 14 patients who were eventually treated, 156 TCRs (median of 8) specific for a median of 4 mutations were detected. The neoantigen-specific TCR sequences expressed by the captured neoantigen-specific T cells were then cloned and functionally characterized in healthy donor T cells for adoptive T cell transfer product selection. Confirmation of the reconstituted neo-TCR binding to the soluble peptide-HLA complex resulted in 64 out of the 156 TCRs (41%) being confirmed as specific and sufficiently functional. Up to 3 confirmed TCR candidates per patient were selected for the final cell transfer product (based on functionality, binding, and diversity across HLAs). The majority of the 35 infused neoTCRs had relatively low affinity as assessed by IFN- γ effective concentrations. Clinical-grade neoTCR transgenic T cell products were generated using a non-viral precision genome engineering approach. The infused T cells contained between 1.9 and 46.8% of neoTCR expressing T cells. Validation studies demonstrated good correlation of the TCR potency between the TCR inserted into healthy donor cells for selection and the TCRs in the final infused product. neoTCR T cells were found to be polyfunctional (CD107a, IL-2, TNF- α , IFN- γ) and specific. Engraftment of the neoTCR T cells was in the single digit percentage range (per total T cells) per TCR with relatively rapid drop off for most neoTCRs T cells and persistence up to 60 days for some in dose levels 2 and 3 (which had higher numbers of infused product and higher intensity lymphodepletion chemotherapy regimens compared to dose level 1). Infused NeoTCR transgenic T cells obtained from the blood of patients after transfusion had similar phenotypes compared to the infused product (mainly Tmemory stem cell and Teffector type). Clonal mutations encoding neoantigens targeted by the infused neoTCR transgenic T cells were largely consistent between tumor biopsies obtained for neoantigen target selection and the repeat biopsies obtained prior to treatment. The majority of TCRs were detected in

post treatment tumor biopsy of a subset of patients. None of the patients had an objective response to the treatment.

The manuscript describes an impressive set of technological innovations demonstrating, for the first time, that it is feasible in the clinic to identify neoantigen-specific TCRs from T cells in the peripheral blood of cancer patients, engineer them into autologous T cells, expand those T cells in vitro and adoptively transfer them back into the patients. This personalized approach can in theory be applied to any cancer patient given that it covers a wide spectrum of HLAs and is thus not restricted to a single antigen in a specific MHC context. Comprehensive analyses assessing and validating the selected neoTCRs and the engineered neoTCR transgenic T cells of the infusion products are presented. The authors also performed comprehensive and in depth analyses of the success rates for peripheral engraftment, the mutational landscape in tumors (and blood for some cases) over time and the presence of neoTCRs in serial pre- and post-treatment tumor biopsies.

The study is highly innovative and of keen interest to the field of cancer immunotherapy. The approach is a true first and the authors should be lauded to conduct such a sophisticated study with the potential to move the needle in this field. Neoantigens have been shown to be highly relevant tumor antigens and current adoptive T cell approaches are limited by the availability of suitable antigens to target.

From a pure clinical perspective, the results are disappointing as no anti-tumor activity was seen in the study. Of note, the information on clinical efficacy is somewhat buried in the manuscript as there is only one sentence in the main text stating that “4 patients had stable disease” (without mentioning that the remaining 10 patients had progression of disease)

Given that the authors ultimately report on the results of a clinical trial, I suggest that a better way to build the narrative (or at least the discussion) would be to make this clear to the reader and provide a more concrete assessment of what from their perspective the main reasons for this lack of efficacy are and what the concrete next steps would be to improve. There are many aspects that seem to have the potential for optimization including:

- Engraftment in the peripheral blood

- o The persistence of the infused neoTCR transgenic T cells in the peripheral blood is modest and was only monitored for up to 60 days. Frequencies decrease substantially after an initial peak, with many neoTCRs being non-detectable after just a few weeks. Are longer term studies being conducted/planned to assess whether there is persistence beyond 60 days? Are higher numbers of neoTCR transgenic T cells needed

- Trafficking into tumors

- o It is encouraging that the majority of neoTCR transgenic T cells were detectable post transfusion by CDR3 sequence quantification (most of which were not present in the tumor prior to treatment). It is also reassuring that the neoTCRs could be directly visualized by FISH and that they were found in close proximity to tumor cells in many instances. However, the neoTCRs in most cases were far outnumbered by other TCRs which may or may not be surprising given the high numbers of neoTCR transgenic T cells transfused. The authors should comment on that.

- Quality and quantity of the transfused neoTCR transgenic T cells

- o A wide range of both numbers and affinity of neoTCRs in the transfused product was seen. TCR

affinity for example ranged from 0.4 pg/ml to 362 pg/ml. Is there room for optimization here? Is there a benchmark for TCR affinity that would be the minimum threshold? How does the TCR affinity relate to the MART and gp100transgenic T cells that the Rosenberg group used achieving clinical responses in melanoma patients. Some context and perspective with what has previously been done in the field should be provided.

The authors should also comment on the feasibility of this approach and on potential strategies to improve.

- While theoretically applicable to the majority of cancer patients given the HLA diversity and “search” rather than “selection” of the target antigens with this personalized adoptive transfer approach, the majority of patients who were enrolled (let alone the ones who signed consent i.e. 187 or underwent TCR screening i.e. 88) ultimately did not get treated.
- The timelines between screening and dosing should be shown for the 14 patients who were eventually treated

Referee #2 (Remarks to the Author):

Foy et al. report results of a first-in-human phase 1 clinical trial designed to validate the feasibility and safety of adoptive T cell therapy with engineered lymphocytes specific for up to 3 patients' specific Neo-antigens.

The approach is based on a pipeline to identify neo-Ags specific TCR from patients's circulating cells, and on CRISPR/Cas9 non-viral genome editing. NeoTCR transgenic cell products result from the simultaneous knock-out of the two endogenous TCR genes, combined with targeted integration of the Neo-Ag specific TCR genes in the TRAC locus.

The approach is innovative, well designed and well described and it proves the principle that patient's specific engineered T cells can be produced for clinical application and safely administered. Some limitations need to be addressed.

Specific comments:

1. Although feasible in principle, the approach is highly demanding. The authors started from 187 screened patients and infused products to 14 patients. This limitation and possible solutions should be discussed. Furthermore, to fully evaluate the feasibility of the approach, a precise timeline should be provided, possibly implemented in Figure 1A.
2. Although the safety is demonstrated, with a single patient experiencing CRS, and most patients' experiencing neutropenia, secondary to conditioning regimens, signs of efficacy are less convincing, and apparently not dependent from the dose of cell infused, nor from the in vivo expansion of infused cells. The significance of stable disease for up to 56 days (4 months in 1 patient) in a cohort of patients with different cancer types, is of difficult interpretation. This issue should also be clarified and discussed.
3. A major limitation of the approach relies on the scanty characterization of the targeted antigens. TCRs to be used in each patient are prioritized based on their functional avidity. Although the

antigens originate from cancer cell mutations found in each patient, no attempts to prioritize antigens based on i. the expression of the restricting HLA molecule by the cancer cells of each patient, ii. the oncogenicity of the targeted molecule, iii. the % of cancer cells harboring the mutation, iv. the effective processing and presentation of the peptide by cancer cells is described. All these investigations could have guided a different choice of TCR and maybe increased the rate of clinical responses.

4. It would be interesting to see correlations between T cell expansion/clinical responses and quality of the TCRs used (ie: CD8 dependency, avidity, antigen profile according to the qualities mentioned in comment 3).

5. The efficiency of TCR redirection in the infused cell product is very low, often <20% (in some cases 1.9%). What are the plans to improve it?

6. More details on the quality of the manufactured product should be provided. What is the efficiency of TCR KO in the final cell product?

Minor comments:

Extended figure 5°: add color legend to the figure

The first report on genome editing to redirect T cell specificity (DOI: 10.1038/nm.2700) should be referenced

The authors state that the expression of the transgenic TCR under the control of the physiologic TCR promoter is advantageous, as shown in previous studies (refs n.17-18). However, controversial data have been published on this issues. Recently it has been reported that the use of a strong promoter provides functional advantages in engineered T cells (DOI: 10.1126/scitranslmed.abg8027). The authors should address this topic and modify the text accordingly.

Which version of the netMHC tool has been employed for the analysis? Please add this info to the material and methods section

Figure 2A: Not only the variable alpha region has been added in the TRAC locus, also a portion of the constant alpha region, as clearly stated in materials and methods. Please correct the figure.

Could the authors clarify why the neoTCRs in the post infusion analysis lacked full length CDR3 sequences?

Figure 4: what is the meaning of the sentence "TCRs lacking barcode confirmation"? Maybe that the codon optimized constant region could not be detected? Please clarify.

Referee #3 (Remarks to the Author):

Foy et al. describe a clinical approach to adoptive TCR cell therapy targeting personalized neoantigens in solid cancers.

The multistep approach involves

- Identification of expressed mutations by next-generation sequencing of patients' tumor samples
- Cloning of antigen-specific TCRs patients T cells using recombinantly expressed HLA/peptide complexes representing potential class I mutant neoantigens
- CRISPR/CAS-assisted, homologous recombination-driven, non-viral transfer of neoantigen-specific TCRs into patient autologous T cells

- Expansion of transgenic T cells and adoptive transfer for clinical immunotherapy in lympho-depleted patients

The authors demonstrate the feasibility and safety of this complex approach in the context of a multicenter phase I dose-escalation clinical trial in which 14 patients (out of 88 screened for TCRs) were successfully treated with transgenic T cell products containing up to three different neoantigen-specific TCRs.

The authors followed the adoptively transferred T cells through a variety of independent assays and demonstrated expansion of the transgenic T cells in vivo as well as evidence of infiltration of TCR-transgenic T cells into tumor biopsies from patients.

The work described by the authors is impressive in terms of the portfolio of individual technologies that were combined to implement the concept of individualized TCR immunotherapies. The technical methods are sound, and the results are well described and compelling. The authors demonstrate the feasibility of their approach to identify neoantigenic TCRs targeting various HLA restrictions. The TCRs are characterized in terms of their specificity and functional avidity. The study merits publication provided that a number of important aspects are addressed:

major issues:

- 1) The characterization of the identified TCRs lacks an assay demonstrating recognition and killing of autologous tumor cells is conferred by neo-TCR. At least for one patient such functional data should be provided.
- 2) The authors should provide a graph showing, at the level of a single patient, the product turnaround time from the time the patient biopsy was taken. The diagram should include information on the time required for sequencing, TCR identification, characterization, preparation of the vector for homologous recombination, preparation of the cell product, release and transfer.
- 3) The authors should provide spider blots visualising the size of the target lesion in treated patients.
- 4) TCR and neoantigen-sequence information is missing. Authors should include a table with the ID of the treated patient, the TCR-CDR3 sequences identified, their HLA specificity, the minimal neo-epitope detected, and affinity information. This information is essential to ensure that respective TCRs can be cloned and findings can be reproduced by the scientific community.
- 5) A weakness of the work is that it is unclear whether TCRs identified in this way can cause tumor regression. Therefore, the authors should add a caveat to their work that mere identification of a TCR by this approach is not evidence that the particular TCR is clinically relevant.

Minor:

Extended Data Table 1 and 2: Please provide tumor entity as information.

Author Rebuttals to Initial Comments:

Reviewer #1 (Remarks to Authors):

From a pure clinical perspective, the results are disappointing as no anti-tumor activity was seen in the study. Of note, the information on clinical efficacy is somewhat buried in the manuscript as there is only one sentence in the main text stating that “4 patients had stable disease” (without mentioning that the remaining 10 patients had progression of disease).

Response: We have expanded the section describing clinical results with the response assessment of the 16 patients, and we provide a new figure panel (Extended Data Figure 8) with a spider plot of the sum of the maximum diameters of target lesions during the trial.

Given that the authors ultimately report on the results of a clinical trial, I suggest that a better way to build the narrative (or at least the discussion) would be to make this clear to the reader and provide a more concrete assessment of what from their perspective the main reasons for this lack of efficacy are and what the concrete next steps would be to improve. There are many aspects that seem to have the potential for optimization including:

- Engraftment in the peripheral blood

o The persistence of the infused neoTCR transgenic T cells in the peripheral blood is modest and was only monitored for up to 60 days. Frequencies decrease substantially after an initial peak, with many neoTCRs being non-detectable after just a few weeks. Are longer term studies being conducted/planned to assess whether there is persistence beyond 60 days? Are higher numbers of neoTCR transgenic T cells needed

Response: We agree with the Reviewer that the neoTCR transgenic T cell persistence was limited, with only two patient samples (0010 and 0613) available beyond 60 days (day 90 and day 106, respectively). For patient 0613, the percentage of neoTCR+ cells (1.3%) was added to the figure legend. For patient 0010, the percentage of neoTCR+ cells was 0.08%, below the LOD of 0.16%. We do not have any additional samples from any of the dosed patients, so longer term persistence will not be able to be further evaluated for this trial.

We also agree that expansion post-infusion was limited, in particular in the patients in the first two dose levels. The *in vivo* expansion improved with the higher dose level, the addition of IL-2, and the improvements in the manufacturing and non-viral gene editing. This is clear with the inclusion of the new data from the two additional patients with improved manufacture that received higher numbers of neoTCR transgenic T cells and had higher *in vivo* expansion. We want to point out that this dose escalation study started with cell doses that may be lower than would be needed for the potential of a clinical response, in particular if we consider that we divided the total cell dose with up to three TCRs in many patients. In the solid tumor setting, higher numbers of adoptively transferred TCR-transgenic cells are likely to be needed as compared to CAR therapies in hematologic cancers. TCR-engineered T cell clinical trials conducted by others have shown clinical activity in the $1-10 \times 10^9$ per TCR range, with no clear dose response beyond 10 billion cells per TCR. However, the targeted TAA or TSA in these studies are expressed at much higher levels than the average neoepitope we targeted in this trial. We have now expanded on this point in the Discussion.

- Trafficking into tumors

o It is encouraging that the majority of neoTCR transgenic T cells were detectable post transfusion by CDR3 sequence quantification (most of which were not present in the tumor prior to treatment). It is also reassuring that the neoTCRs could be directly visualized by FISH and that they were found in close proximity to tumor cells in many instances. However, the neoTCRs in most cases were far outnumbered by other TCRs which may or may not be

surprising given the high numbers of neoTCR transgenic T cells transfused. The authors should comment on that.

Response: We agree that neoTCRs were outnumbered by the WT TCRs as measured by CDR3 sequences, particularly in the patients enrolled in dose levels 1 and 2 shown in Figure 4b. We observe much higher levels of neoTCR CDR3 reads (all three neoTCRs falling within the top 10 TCRs detected) particularly for the two patients treated in dose level 3. We now include this point in the expanded Discussion. Higher dose and higher percentage of neoTCR in the final cell product resulted in further increased infiltration.

- Quality and quantity of the transfused neoTCR transgenic T cells
o A wide range of both numbers and affinity of neoTCRs in the transfused product was seen. TCR affinity for example ranged from 0.4 pg/ml to 362 pg/ml. Is there room for optimization here? Is there a benchmark for TCR affinity that would be the minimum threshold? How does the TCR affinity relate to the MART and gp100transgenic T cells that the Rosenberg group used achieving clinical responses in melanoma patients. Some context and perspective with what has previously been done in the field should be provided.

Response: We agree with the Reviewer that our trial included neoTCRs with a large affinity range. This is because we initially casted a wide net based on literature that had suggested that low affinity TCRs T cells could be beneficial in chronic viral infections. As more clinical data became available in 2020 and 2021, we tightened IFN γ EC₅₀ criteria for product selection for the highest affinity TCRs. Furthermore, we retrospectively compared the IFN γ EC₅₀ of the clinically infused TCRs (NeoTCR-P1) to several clinically active TCRs that we generated in healthy donor T cells (new Extended Data Figure 1c). Half (18/37) of the neoTCRs isolated and infused into patients had a similar TCR affinity to those that have seen activity in the clinic. We have included this point in the Results and Discussion sections. Please note that the data below was generated using T cells from healthy donor cells at product selection, and is reported from the Cytokine Bead Array in ng/mL, whereas the data from lot release in the question is measured by ELISA and reported as pg/ml.

The authors should also comment on the feasibility of this approach and on potential strategies to improve.

- While theoretically applicable to the majority of cancer patients given the HLA diversity and “search” rather than “selection” of the target antigens with this personalized adoptive transfer

approach, the majority of patients who were enrolled (let alone the ones who signed consent i.e. 187 or underwent TCR screening i.e. 88) ultimately did not get treated.

Response: We have included additional comments on the high drop-out rate observed at screening though TCR selection noted in the Consort diagram, particularly a result of sample quality. We have also added our thoughts on potential strategies to improve the personalized approach in the Discussion (paragraph 5), including the potential to optimize the cell therapy and making the process shorter using a pre-established library of TCRs to common mutations and viral antigens with multiple HLA specificities selected using the same approach as in this clinical trial.

- The timelines between screening and dosing should be shown for the 14 patients who were eventually treated

Response: We include a new figure panel (Extended Data Figure 4b) with a swimmer's lane plot that details for each of the 16 patients the different periods of the trial, from consenting, screening, sample acquisition and sequencing, neoTCR isolation, neoTCR selection, T cell engineering and cellular product reinfusion to the patients.

Reviewer #2 (Remarks to Authors):

Specific comments:

1. Although feasible in principle, the approach is highly demanding. The authors started from 187 screened patients and infused products to 14 patients. This limitation and possible solutions should be discussed. Furthermore, to fully evaluate the feasibility of the approach, a precise timeline should be provided, possibly implemented in Figure 1A.

Response: As requested by the three Reviewers, in the resubmission we include a specific timeline of patients in (Extended Data Figure 4b). In addition, we have expanded the Discussion paragraph to discuss the limitations of the approach with personalized neoTCRs for each patient, and potential solutions using a pre-established library of TCRs to common mutations and viral antigens with multiple HLA specificities selected using the same approach as in this clinical trial.

2. Although the safety is demonstrated, with a single patient experiencing CRS, and most patients' experiencing neutropenia, secondary to conditioning regimens, signs of efficacy are less convincing, and apparently not dependent from the dose of cell infused, nor from the *in vivo* expansion of infused cells. The significance of stable disease for up to 56 days (4 months in 1 patient) in a cohort of patients with different cancer types, is of difficult interpretation. This issue should also be clarified and discussed.

Response: In the resubmission we have included a new section in the Results that expands on the clinical responses and patient outcomes. We agree that the significance of disease stabilization is difficult to interpret, in particular in the context of a therapy that includes high dose conditioning chemotherapy. In this regard, the last patient in the series that is added to the resubmission had decrease in size of two metastatic lesions (Extended Data Figure 8b), with progression at other sites. This suggests to us that it was not a response to chemotherapy (should be the same at all sites), in particular as this patient had a cancer with low potential of response to chemotherapy (lung cancer), and the patient had received 6 prior lines of therapy including chemotherapy.

3. A major limitation of the approach relies on the scanty characterization of the targeted antigens. TCRs to be used in each patient are prioritized based on their functional avidity. Although the antigens originate from cancer cell mutations found in each patient, no attempts to prioritize antigens based on i. the expression of the restricting HLA molecule by the cancer cells of each patient, ii. the oncogenicity of the targeted molecule, iii. the % of cancer cells harboring the mutation, iv. the effective processing and presentation of the peptide by cancer cells is described. All these investigations could have guided a different choice of TCR and maybe increased the rate of clinical responses.

Response: The Reviewer provides very relevant input and many of these suggestions were updated for the last 2 patients dosed. i) The expression levels of the restricting HLA molecule were available in version 3 of the pipeline and considered at time of product selection. ii) Priority bioinformatics neoantigen mutation selection was given to cancer driver mutations for the last 2 dosed patients and was always assessed at time of product selection for all patients iii) Truncality and cancer cellular fraction was predicted for every neoantigen and always considered for all samples at time of product selection with priority always given to truncal neoantigens. iv) Although this data would be of high value, this step would have extended the turnaround time even further and was outside the scope of this clinical trial. These details have been made clearer in the methods description.

4. It would be interesting to see correlations between T cell expansion/clinical responses and quality of the TCRs used (ie: CD8 dependency, avidity, antigen profile according to the qualities mentioned in comment 3).

Response: Our best responses in this phase 1 trial were stable disease. In the absence of objective clinical responses, it is difficult to draw meaningful conclusions here with respect to quality of the TCRs and clinical response. However, we agree that it is still interesting to see if characteristics of the TCR affect other measures of expansion and trafficking. We looked for several correlations between the quality (phenotype, CD8-dependency, TCR affinity) of the final cell product and *in vivo* expansion as measured by peripheral blood neoTCR-engineered T cell pharmacokinetics, and found that the percentage of neoTCR positive cells detected in the final cell product was strongly correlated with the PK area under the curve (day +7, initial expansion). This is included in Extended Data Figure 4f. Further analysis of the tumor biopsy shows increased infiltration with lower IFN γ EC₅₀ or higher TCR affinity (a measure of CD8-dependency). We have added an additional Figure 7d, based on this comment.

5. The efficiency of TCR redirection in the infused cell product is very low, often <20% (in some cases 1.9%). What are the plans to improve it?

Response: We have worked on improving the neoTCR gene editing efficiency with updates to the manufacturing process including changing the media (v 2.1) and using a new electroporation device (v 3.0). We demonstrate that it is feasible at clinical grade with the last two patients that were added in this resubmission. The implementation of the new device improved both the knock-in and knock-out efficiencies, which significantly reduced the percent of wild type cells in the final cell product (Extended Data Figure 3c).

6. More details on the quality of the manufactured product should be provided. What is the efficiency of TCR KO in the final cell product?

Response: Using the manufacturing versions 2.0 and 2.1, KO efficiency was around 35% - 48% (total TCR KO + neoTCR KO/KI). This was significantly improved the implementation of the new electroporation device. Using manufacturing version 3, this increased to 87% for the two patients (three TCRs) evaluated for the final cell product (see updated Extended Data Figure 3c, above).

Minor

comments:

Extended figure 5°: add color legend to the figure

Response: Thank you for pointing this out, we have now added the legend.

The first report on genome editing to redirect T cell specificity (DOI: 10.1038/nm.2700) should be referenced

Response: We thank the Reviewer for bringing up this relevant reference, which is now added to the article.

The authors state that the expression of the transgenic TCR under the control of the physiologic TCR promoter is advantageous, as shown in previous studies (refs n.17-18). However, controversial data have been published on this issues. Recently it has been reported that the use of a strong promoter provides functional advantages in engineered T cells (DOI: 10.1126/scitranslmed.abg8027). The authors should address this topic and modify the text accordingly.

Response: We thank the Reviewer for bringing up this relevant point and reference, which is now added to the article.

Which version of the netMHC tool has been employed for the analysis? Please add this info to the material and methods section

Response: We confirm that we used netMHCpan 3.0, netMHCpan4.0, and netMHCpan4.1, and we added this information to the Material and Methods section.

Figure 2A: Not only the variable alpha region has been added in the TRAC locus, also a portion of the constant alpha region, as clearly stated in materials and methods. Please correct the figure.

Response: We have updated Figure 2a to address this issue.

Could the authors clarify why the neoTCRs in the post infusion analysis lacked full length CDR3 sequences?

Response: Thank you to the reviewer for highlighting this sentence as it can be clarified further. The neoTCR did not lack the full length CDR3 sequence, but instead it could not be detected by *sequencing* which may be due to lack of efficient trafficking guided by that particular neoTCR. We have now clarified this in the text. We also added and reference Extended Figure 7c to help readers visualize reads mapped to the CDR3 and flanking barcode sequence.

The sentence was clarified: “For patient 0503 infused with a three neoTCR product, the presence of the neoTCRs were inferred by the flanking barcode sequence (Figure 4a and b) but the CDR3 sequence for the specific neoTCR could not be resolved.”

Figure 4: what is the meaning of the sentence “TCRs lacking barcode confirmation”? Maybe that the codon optimized constant region could not be detected? Please clarify.

Response: The sentence was clarified-

Colours of boxes indicate cancer type and one TCR for which the sequencing reads supporting the codon optimized constant region was not detected is marked with an orange line (patient 0010). Additionally, the referenced Extended Data Figure 7c should also help to clarify this further.

Referee #3 (Remarks to the Author):

Major issues:

1) The characterization of the identified TCRs lacks an assay demonstrating recognition and killing of autologous tumor cells is conferred by neo-TCR. At least for one patient such functional data should be provided.

Response: In the resubmission we have added Extended Data Figure 1b to directly address the Reviewer's comment. This new figure describes the functional validation of the antitumor activity of a TCR isolated from the blood of a patient with colon cancer using the PACT neoTCR isolation approach, the cloning of this neoTCR and non-viral gene editing to have it express by T cells, and then co-culture of these T cells with an HLA-matched cell line that was precision genome engineered to carry the cognate mutation (R20Q) on either a single or both alleles of the COX6C gene, resulting in endogenous expression levels of the neoE. We believe that this experiment provides a valid example of the functionality of the neoTCRs isolated for this trial, and it is included in the article so it can be reviewed independent of other information.

In addition, as anticipated by the Reviewer, we had put a lot of work to functionally validate the PACT neoTCR isolation and reconstitution approach to make sure that the neoTCRs we infused to patients had the characteristics that we would desire to recognize and kill patient-matched cancer cells. This could not be done directly with the samples from the patients in the trial, as establishing cancer cell lines is a lengthy process with a very low yield, in particular from some of the indications like colorectal cancer, which was the majority of the referrals for the trial in the early phases. Therefore, we provide a whole separate article for the Reviewers, which we are finalizing to submit to Nature for consideration. In this article, we describe the isolation of neoTCRs from six patients with metastatic melanoma for which we had established a patient-specific melanoma cell line, and we used the PACT neoTCR isolation approach to study differential responses to anti-PD-1-based immunotherapy. There were three patients with clinical response and three without a clinical response to anti-PD-1-based therapy. In total, we isolated 62 neoTCRs from these six patients, and they all showed ability to recognize the patient-specific melanoma cell line when neoTCR engineered T cells were cocultured with the patient-matched melanoma cell line in terms of specific cytokine production (interferon-gamma, TNF, IL-2) and T cell activation markers (4-1BB). 39 of the 62 TCRs also induced

specific cytotoxicity against the patient-matched melanoma cell lines. The main data is included in Figures 3 and 4 of the Puig-Saus et al. article. We hope that this additional information helps the Reviewers evaluate the amount of work we had conducted to optimize the process to isolate functional neoTCRs to reinfuse to these 16 patients. Part of the data from this article had been presented orally at a 2021 AACR Next Generation session (Puig-Saus et al.), and we are citing the abstract of that presentation in the resubmitted article.

2) The authors should provide a graph showing, at the level of a single patient, the product turnaround time from the time the patient biopsy was taken. The diagram should include information on the time required for sequencing, TCR identification, characterization, preparation of the vector for homologous recombination, preparation of the cell product, release and transfer.

Response: We thank the Reviewer for this suggestion, and we have now included such a graph in Extended Data Figure 4b. This is a figure using a swimmer's lane kind of plot that provides individual information on the timeline of each of the 16 patients enrolled in the trial, and how certain periods were extended for some of the patients.

3) The authors should provide spider blots visualizing the size of the target lesion in treated patients.

Response: As also requested by the other Reviewers, we expanded the presentation of clinical results in the article, and we now include the spider plot of the sum of maximum diameters of target lesions for all the patients with evaluable disease as requested by Reviewer #3 (Extended Data Figure 8a).

4) TCR and neoantigen-sequence information is missing. Authors should include a table with the ID of the treated patient, the TCR-CDR3 sequences identified, their HLA specificity, the minimal neo-epitope detected, and affinity information. This information is essential to ensure that respective TCRs can be cloned and findings can be reproduced by the scientific community.

Response: We agree with the Reviewer that such a table would be a useful resource for the field, and we now include Supplemental Information Table 2 with the TCR alpha and beta gene sequences, HLAs and targeted mutations.

5) A weakness of the work is that it is unclear whether TCRs identified in this way can cause tumor regression. Therefore, the authors should add a caveat to their work that mere identification of a TCR by this approach is not evidence that the particular TCR is clinically relevant.

Response: We have included this caveat in the Discussion.

Minor:

Extended Data Table 1 and 2: Please provide tumor entity as information.

Response: The code of each patient is the same for Extended Data Tables 1, 2 and 3, with the information about the patient characteristics (age, cancer histology, number of prior lines of therapy) included in Extended Data Table 3. We just noted that the title of this table in the index page of the Extended Data was incorrect, and we have corrected it in the resubmission.

Reviewer Reports on the First Revision:

Referees' comments:

Referee #1 (Remarks to the Author):

The authors have addressed all my comments. I congratulate them to their success in applying this highly innovative and complex technology in cancer patients - a true conceptual advance. While demonstrating that a fully personalized neoantigen TCR ACT approach might be too cumbersome (yet) in the clinic, the comprehensive assessments of the individual cell products and genomic/immunologic analyses provide a clear roadmap for further innovation and improvements.

Referee #2 (Remarks to the Author):

The authors have addressed all issues raised.